# Human FCHO1 deficiency reveals role for clathrin-mediated endocytosis in development and function of T cells

Marcin Łyszkiewicz[1,12,13✉], Natalia Ziętara[1,12,13], Laura Frey[1], Ulrich Pannicke[2], Marcel Stern[3], Yanshan Liu[1], Yanxin Fan[1], Jacek Puchałka[1,14], Sebastian Hollizeck[1], Ido Somekh[1], Meino Rohlfs[1], Tuğba Yilmaz[4], Ekrem Ünal[4], Musa Karakukcu[4], Türkan Patiroğlu[4,5], Christina Kellerer[2], Ebru Karasu[2], Karl-Walter Sykora[6], Atar Lev[7], Amos Simon[7], Raz Somech[7], Joachim Roesler[8], Manfred Hoenig[9], Oliver T. Keppler[3,10], Klaus Schwarz[2,11] & Christoph Klein[1✉]

Clathrin-mediated endocytosis (CME) is critical for internalisation of molecules across cell membranes. The FCH domain only 1 (FCHO1) protein is key molecule involved in the early stages of CME formation. The consequences of mutations in *FCHO1* in humans were unknown. We identify ten unrelated patients with variable T and B cell lymphopenia, who are homozygous for six distinct mutations in *FCHO1*. We demonstrate that these mutations either lead to mislocalisation of the protein or prevent its interaction with binding partners. Live-cell imaging of cells expressing mutant variants of FCHO1 provide evidence of impaired formation of clathrin coated pits (CCP). Patient T cells are unresponsive to T cell receptor (TCR) triggering. Internalisation of the TCR receptor is severely perturbed in FCHO1-deficient Jurkat T cells but can be rescued by expression of wild-type FCHO1. Thus, we discovered a previously unrecognised critical role of FCHO1 and CME during T-cell development and function in humans.

[1] Department of Pediatrics, Dr. von Hauner Children's Hospital, University Hospital, LMU, Munich, Germany. [2] Institute for Transfusion Medicine, University of Ulm, Ulm, Germany. [3] Max von Pettenkofer Institute, Virology, National Reference Center for Retroviruses, Faculty of Medicine, LMU München, Munich, Germany. [4] Department of Pediatrics, Division of Pediatric Hematology & Oncology, Erciyes University, Kayseri, Turkey. [5] Department of Pediatrics, Division of Pediatric Immunology, Erciyes University, Kayseri, Turkey. [6] Department of Pediatric Hematology/Oncology, Hannover Medical School, Hannover, Germany. [7] Pediatric Department A and the Immunology Service, Jeffrey Modell Foundation Center, Edmond and Lily Safra Children's Hospital, Sheba Medical Center, Tel Hashomer and Sackler Faculty of Medicine Tel Aviv University, Tel Aviv, Israel. [8] Department of Pediatrics, Carl Gustav Carus Technical University Dresden, Dresden, Germany. [9] Department of Pediatrics, University Medical Centre Ulm, Ulm, Germany. [10] German Center for Infection Research (DZIF), Partner Site Munich, Munich, Germany. [11] Institute for Clinical Transfusion Medicine and Immunogenetics Ulm, German Red Cross Blood Service Baden-Wuerttemberg, Hessen, Germany. [12] Present address: Institute for Immunology, Biomedical Center Munich, Ludwig-Maximilians-Universität München, Planegg-Martinsried, 82152 Munich, Germany. [13] These authors contributed equally: Marcin Łyszkiewicz, Natalia Ziętara. [14] Deceased: Jacek Puchałka. ✉email: Marcin.Lyszkiewicz@med.uni-muenchen.de; Christoph.Klein@med.uni-muenchen.de

Eukaryotic cells are characterised by a structural and functional compartmentalisation. Biological membrane systems are critically involved in the establishment and maintenance of the cellular compartments. The highly dynamic modulation of membranes is a key feature for their role in a variety of biological processes, such as endocytosis, secretion, signal transduction or migration. Clathrin, a central molecular scaffold, plays an essential role in the re-organisation of cellular membranes and the transport between compartments. Clathrin-coated vesicles (CCV), first described as "clathrin-coated pits" (CCP) in mosquito oocytes[1] and later purified from pig brain[2], control internalisation of membrane-associated proteins and protein transport from the trans-Golgi-network[3].

Clathrin-dependent endocytosis (CME) is initiated at the plasma membrane by the recruitment of adaptors (heterotetrameric AP-2 complex, AP180; clathrin assembly lymphoid myeloid leukaemia, CALM; and anchor proteins (FCH domain only 1 and 2, [FCHO1 and 2]; epidermal growth factor receptor substrate 15, [EPS15]; and intersectin)[4–6]. These factors are enriched within PIP2 (phosphatidylinositol 4,5,-biphosphate)-rich regions of the membrane and trigger the assembly of the clathrin proteins into clusters[7]. During maturation, the clathrin-enriched membrane patches undergo a local curvature and an invaginated spheric clathrin-coated pit is generated. The connection of this pit to the plasma membrane is detached by dynamin[8].

The molecular details of the formation of a new vesicle are still not fully understood. In vitro, the presence of clathrin, an adaptor protein (i.e., AP-1, AP-2) and dynamin are sufficient to form a vesicle[9]. However, in vivo, the mechanism is much more complex. It has been shown by live-cell total internal reflection fluorescence (TIRF) imaging with single-molecule EGFP sensitivity and high-temporal resolution that the formation of a vesicle can originate from two AP-2 molecules with a clathrin triskeleton[4]. In contrast, another report suggested that FCHO1 and FCHO2 need to be associated with the membrane prior to AP-2 recruitment to initiate the process[5]. FCHO1/2 interact and show similar kinetics at the membrane as the scaffold proteins EPS15 and intersectin, two factors that are important for clathrin-dependent endocytosis[6,10]. Upon downregulation of EPS15 expression by small-interfering RNA, FCHO1/2 show a diffuse pattern of distribution at the cell membrane. This observation suggests that EPS15 is critical for the specific agglomeration of FCHO1/2 on the site of the formation of clathrin-coated vesicles[5].

Since the early clinical discoveries of monogenic immune disorders by Rolf Kostmann[11] and Ogden Bruton[12], >400 primary immunodeficiency diseases have been delineated[13], many of which have opened unprecedented insights into molecular mechanisms orchestrating differentiation and function of the human immune system.

Here, we describe ten human patients with T-cell deficiency and loss-of-function mutations in FCHO1. Our experiments demonstrate that the absence of functional FCHO1 results in perturbed clathrin-mediated endocytosis in several tissues, as well as dysfunctional internalisation of the TCR. Pharmacological inhibition of CME during in vitro T-cell development results in marked delay of T-cell differentiation. In summary, we identify an essential and non-redundant role for the clathrin adaptor FCHO1 in T-cell differentiation and function, linking CME to the function of the human immune system.

## Results

**Clinical phenotype and molecular genetics.** We collected seven pedigrees with ten patients presenting with features of T-cell immunodeficiency (Fig. 1a and Supplementary Fig. 1). All patients suffered from severe bacterial, viral or fungal infections indicative of a primary immunodeficiency disorder. Three patients (B1, C1, D1) developed B-cell lymphoma prior to allogeneic hematopoietic stem cell transplantation, and three patients (A1, D1, E1) had neurological disease (Table 1). Immunological parameters were ranging from a moderate decrease in peripheral CD4+ T cells (F2 and G1) to severe combined immunodeficiency with virtually absent B- and T-cells (A1, B1, E1). With the exception of G1, all patients had hypogammaglobulinemia (Supplementary Table 1). Thus, the common immunophenotypic denominator for all these patients was CD4+ T-cell deficiency.

In order to identify the underlying genetic defect, we performed whole-exome sequencing (WES) followed by Sanger sequencing of candidate genes on patients and family members (see Online Methods). We have identified six distinct, novel and segregating homozygous mutations in FCHO1 in seven pedigrees (Fig. 1b and Supplementary Figs. 2 and 3; Table 1 and Supplementary Table 2). At the DNA level, the mutation in kindred A results in a nucleotide substitution at the position c.2036 G > C, in kindred B in a nucleotide substitution at the position c.100 G > C, whereas the mutations of the pedigrees C and D (not known to be connected by kinship) are insertions c.2023insG resulting in a frameshift and premature termination of the protein (Fig. 1b, c). The mutations of kindred E (changing the first nucleotide of intron 8 c.489 + 1 G > A) and kindred F (contained in the intron 6 splice acceptor site; c.195-2 A > C) affect splicing of pre-mRNA (Fig. 1c and Supplementary Fig. 2). Effects on FCHO1 transcripts in fibroblasts, peripheral blood mononuclear cell (PBMC) and EBV-LCL are shown in Supplementary Figs. 2 and 3. No bands corresponding to wild-type FCHO1 were detected in these analyses. In kindred G, a single-nucleotide substitution at position c.1948C > T results in a predicted premature stop codon at amino acid position 650. All identified patient-associated mutations are summarised in Supplementary Table 2.

FCHO1 consists of three main segments organised into two major domains (Fig. 1c). The N-terminal F-BAR domain (also known as extended FCH), which is responsible for membrane binding, is followed by a structurally less organised and evolutionary poorly conserved segment that binds the adaptor protein AP-2. The C-terminal part (~270 amino acids) forms a μ homology domain (μHD), which directly binds to epidermal growth factor receptor pathway substrate 15 (EPS15) and EPS15-like 1 (EPS15L1) also known as EPS15R[5].

The mutation in kindred A results in an Arg to Pro substitution at the position 679 in the μHD domain. The crystal structure of the Danio rerio μHD domain of Fcho1 has been resolved together with a fragment of EPS15 allowing for direct modelling the effect of the amino acid substitution on this structure (Fig. 1d)[14]. Although the patient-associated mutations are not directly located at the protein-binding trough, replacing the charged side chain of Arg with a nonpolar and rigid ring of Pro may result in steric alterations in the μHD subdomain A and thus affect interaction with its binding partners (Fig. 1d).

The point mutation in kindred B results in an amino acid substitution in the alpha-helix structure of the F-BAR domain (p. Ala34Pro). The F-BAR domain of FCHO1 has not been crystallised. However, the structure of its functional and structural paralogue FCHO2 has been resolved, which allows for modelling of the patient-associated mutation[14]. The substitution of Ala with Pro at position 34 is likely to result in a disruption of the alpha-helix structure, which in turn may lead to alterations in the membrane-binding properties of the entire domain (Fig. 1e).

In addition to mutations resulting in amino acid substitutions, we identified three distinct variants resulting in a premature stop

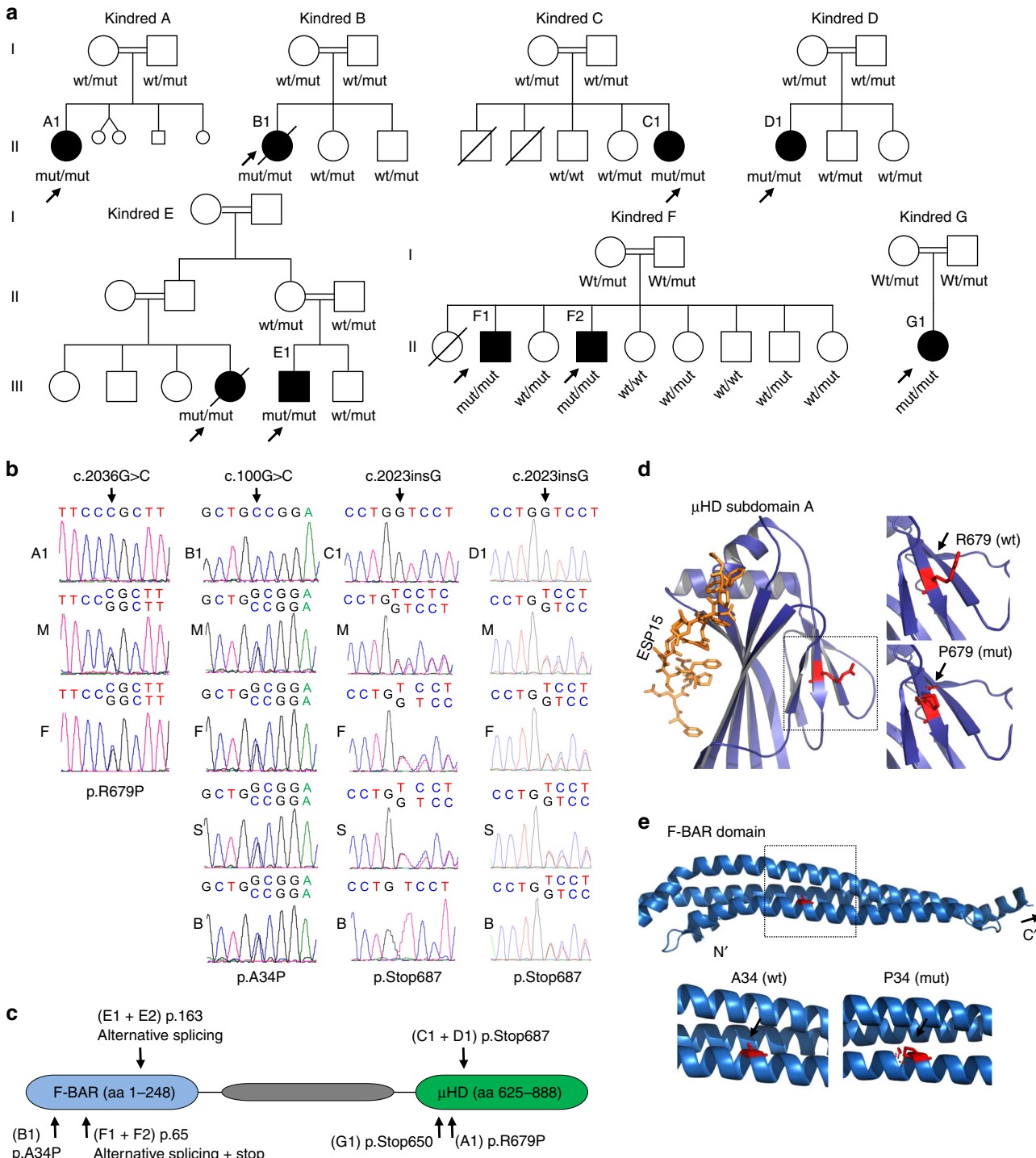

**Fig. 1 Homozygous mutations in the *FCHO1* gene segregate with patients' lymphopenia. a** Pedigrees of seven unrelated families show ancestral segregation of mutations in the *FCHO1* locus. Generations are assigned by Roman numerals from I to III. Index cases are marked with an arrow, small circles and squares denote spontaneous abortions and crossed symbols deceased individuals. mut, mutation; wt, wild-type. **b** Sanger sequencing chromatograms indicating homozygous mutation c.2036 G > C in index cases of kindred A, c.100 G > C in kindred B and c.2023insG in kindred C and D. Families C and D are not connected by kinship. A–D, index cases; M' mother; F, father; S, sister; B, brother. Additional kindred analyses are exhibited in Supplementary Figs. 2 and 3. **c** Schematic representation of FCHO1 protein indicating two main domains and localisation of family-associated mutations. **d**, **e** Computed crystal structures with indicated point mutations in μHD domain (**d**) and F-BAR domain (**e**).

codon. The two identical mutations in the unrelated families C and D result in a frameshift and premature stop codon at amino acid position 687 (p.Val675GlyTer13, referred to as p.Stop687). Similarly, the mutation associated with kindred G results in a premature stop codon at amino acid position 650 (p.Stop650). In

both variants virtually the entire μHD domain is truncated, which presumably alters interaction with the FCHO1-binding partners. The point mutation in kindred E affects a splice donor and thereby results in a shortening of the FCHO1 by 51 amino acid encoded by exon 8 (Supplementary Fig. 2). Given the localisation

**Table 1 Summary of clinical features of patients carrying mutations in *FCHO1*.**

| Patient | Origin | Genetic variant[a] | Consequences of mutation | Immunological findings | Infections | Other clinical findings | Therapy and outcome |
|---|---|---|---|---|---|---|---|
| A1 | Germany | *FCHO1* c.2036 G > C | aa substitution in μHD domain (p.R679P) | • T- and B-cell lymphopenia<br>• hypogammaglo-bulinemia | • Recurrent pneumonia and viral gastroenteritis<br>• Relapsing oro-genital mycoses<br>• Bronchiolitis obliterans<br>• Postpneumonic pulmonary fibrosis<br>• Otitis media | • Moya-Moya syndrome<br>• Transient left hemiparesis upon cerebral ischaemia<br>• Failure to thrive<br>• Microcephaly | Reduced cardiopulmonary performance, stable Moya-Moya 9 years after HLA-matched HSCT |
| B1 | Turkey | *FCHO1* c.100 G > C | aa substitution in F-BAR domain (p.A34P) | • T- and B-cell lymphopenia<br>• hypogammaglo-bulinemia | • Recurrent pneumonia<br>• Recurrent fungal infections<br>• CMV infection | • DLBCL<br>• Renal metastases | Deceased as consequence of DLBCL, age 16 years |
| C1 | Turkey | *FCHO1* c.2023insG | Truncated (p.Stop687) | • CD4+ T-cell lymphopenia<br>• hypogammaglo-bulinemia | • Recurrent pulmonary infections<br>• Recurrent fungal infections<br>• Otitis media | • EBV+ Hodgkin lymphoma<br>• Failure to thrive<br>• hepatosplenomegaly<br>• Renal masses<br>• Xanthogranulomatous pyelonephritis | IVIG replacement and antibiotics; awaiting allo-HSCT |
| D1 | Turkey | *FCHO1* c.2023insG | Truncated (p.Stop687) | • CD4+ T-cell lymphopenia<br>• hypogammaglo-bulinemia | • Recurrent pneumonia<br>• HSV infection | • DLBCL stage IV<br>• Liver lesions<br>• Spleen lesions<br>• Lung lesions<br>• Aphthous stomatitis<br>• Gingivitis<br>• Encephalitis | Deceased, age 10 years |
| E1 | Palestine | *FCHO1* c.489 + 1G > A | Alternative splicing IVS8 splice donor | • CD4+ T-cell and B-lymphopenia<br>• hypogammaglo-bulinemia | • Recurrent pneumonia<br>• Chronic diarrhoea<br>• CMV infection<br>• Fungal infection | • Mild brain atrophy | IVIG replacement and antibiotics; awaiting allo-HSCT |
| E2 | Palestine | *FCHO1* c.489 + 1G > A | Alternative splicing IVS8 splice donor | • not available | • Recurrent pneumonia<br>• Chronic diarrhoea | | Deceased after cardiac arrest, age 2 years |
| E3 | Palestine | *FCHO1* c.489 + 1G > A | Alternative splicing IVS8 splice donor | • CD4+ T-cell lymphopenia<br>• hypogammaglo-bulinemia | • Recurrent pneumonia<br>• Chronic diarrhoea<br>• EBV infection | | IVIG replacement and antibiotics; awaiting allo-HSCT |
| F1 | Saudi Arabia | *FCHO1* c.195-2 A > C | Alternative splicing IVS6 splice acceptor | • CD4+ T-cell lymphopenia<br>• hypogammaglo-bulinemia | • Recurrent pneumonia<br>• Chronic diarrhoea<br>• Cryptosporidiosis<br>• Recurrent stomatitis (HSV) | • Failure to thrive | HSCT at age 5 yrs (no conditioning), MFD (mother), a + cGvHD, complete donor chimerism, normal immune function, off IVIG, 10 yrs follow up |
| F2 | Saudi Arabia | *FCHO1* c.195-2 A > C | Alternative splicing IVS6 splice acceptor | • CD4 + T-cell lymphopenia | • Recurrent pneumonia<br>• Chronic diarrhoea<br>• Cryptosporidiosis<br>• Multiple viruses (adenovirus, RSV, enterovirus) | | HSCT at age 1.5 yrs, (no conditioning), MSD, no GvHD, post-transplant intracranial EBV-PTLD and atypical mycobacterium-associated mastoiditis; mixed chimerism (T-cells 100% donor, non-T-MNCs 5-10% donor, red cells recipient), normal immune function, off IVIG, 12.5 yrs follow up |
| G1 | Algeria | *FCHO1* c.1948C > T | Truncated p.R650X p.Stop650 | • CD4 + T-cell lymphopenia<br>• Weak response to vaccination | • Recurrent broncho-pulmonary infections<br>• Candidiasis<br>• CMV infection | • Failure to thrive | HSCT (MFD) at age 5 years, doing well |

*EBV* Epstein-Barr virus, *DLBCL* diffuse large B-cell lymphoma, *PTLD* post-transplant lymphoproliferative disorder, *HLA* human leucocyte antigen, *HSCT* haematopoietic stem cell transplantation, *MFD* matched family donor, *IVIG* intravenous immunoglobulin, *a* + *cGvHD* acute and chronic graft versus host disease.
[a]Sequence of coding DNA is given from the first nucleotide of the translation start codon.
[b]Sequence of protein is given from the first amino acid.

of these amino acids, it is likely that, in case a protein is stably expressed, it disrupts the alpha-helix structure of the F-BAR domain. Finally, the kindred F-associated mutation in the splice acceptor region gives rise to three possible splice variants, each of them resulting in a premature stop codon shortly downstream of exon 6. It prevents correct splicing and by prediction presumably leads to expression of a very short (~7.1 kDa), non-functional form of FCHO1 (Supplementary Fig. 3b–e).

**Effects of patient-associated mutations on FCHO1 function.** Based on CADD and PolyPhen-2 scores, all patient-associated mutations are deleterious for the FCHO1 protein function (Supplementary Table 2). To test this, we first established a heterologous system, where HEK239T cells were transiently transfected with vectors carrying wild-type (wt) or mutated versions of the *FCHO1* complementay DNA (cDNA). None of the herein tested mutations altered protein stability, albeit p.Stop687 resulted in the expression of a shorter protein (Fig. 2a and Supplementary Fig. 4). The encoded proteins were tested for co-immunoprecipitation with their direct interacting partners EPS15 and its homologue EPS15R. Recovery of the EPS15 and EPS15R was strongly reduced in both mutants affecting the µHD domain (c.R679P and p.Stop687). In contrast, the FCHO1 mutant affecting the F-BAR domain (p.A34P) did not alter the interaction with EPS15 and EPS15R (Fig. 2a).

The function of FCHO1 critically depends not only on its biochemical interaction with partner proteins but also on the spatiotemporal organisation of its interactome. Nucleation of FCHO1-mediated clathrin-coated vesicles (CCV) occurs only at the plasma membrane[5]. We, therefore, set out to test whether the identified *FCHO1* mutations altered the subcellular location of the corresponding protein. We employed SK-MEL-2 cells expressing a RFP-tagged clathrin light chain from one allele of its endogenous locus[15]. In order to circumvent confounding effects of the endogenously encoded FCHO1 protein, the *FCHO1* gene was deleted using CRISPR/Cas9-mediated gene editing. Transient expression of wt FCHO1-GFP fusion protein in knockout (ko) cells showed scattered bright puncta associated with the plasma membrane as previously reported[5,14] (Fig. 2b–d and Supplementary Figs. 5–7). In contrast, expression of the F-BAR mutated (p.A34P) FCHO1 form resulted in the formation of large plasma membrane-dissociated agglomerations (Fig. 2b–d and Supplementary Figs. 5–7).

Further, both pR679P and p.Stop687 µHD mutants failed to form any punctated structures. These mutants appeared as a diffuse network, mostly dissociated from the plasma membrane (Fig. 2b–d and Supplementary Fig. 5). In accordance with the co-immunoprecipitation results, only wt and p.A34P FCHO1 colocalized with their partners EPS15 and adaptin (Fig. 2b–c, and for quantification Fig. 2e). Importantly, all mutants failed to colocalize with endogenous clathrin (Fig. 2d–e). We chose a model system with physiological expression levels of FCHO1. While this is advantageous in many respects[5], it is limited with respect to poor signal-to-noise ratio. To improve signal intensities, we additionally transduced SK-MEL-2 cells to stably over-express RFP-tagged clathrin light chains. As shown in Supplementary Fig 7, these data confirm that only wild-type FCHO1, but none of the mutants, colocalize with clathrin.

Nucleation of CCV is a highly dynamic process. To measure the dynamics of protein–protein interaction between FCHO1 and clathrin, we established live-cell imaging models. Here, we followed wt FCHO1 and both point mutants (p.R679P and p.A34P). As anticipated, wt GFP-FCHO1 initiated the formation of clathrin-coated pits, whereas both mutants failed to do so (Fig. 3a–c and Supplementary Movies 1–3). FCHO1 carrying the

µHD-associated p.R679P mutation prevented the formation of productive pits and thus failed to facilitate the formation of CCV. Since the sensitivity of the confocal microscope does not allow to discriminate single molecules, we cannot exclude that small, abortive pits may have been formed. However, we failed to detect any aggregation of FCHO1 carrying this mutation. In striking contrast, the F-BAR-associated mutation p.A34P resulted in the formation of fast moving, large aggregates of GFP-FCHO1 fusion proteins. Moreover, such membrane-dissociated aggregates tended to merge, leading to the formation of cytoplasmic protein intrusions (Supplementary Fig. 5). These pits appeared to be abortive, since they failed to interact with clathrin (Fig. 3a–c).

These results illustrate that the identified mutations in *FCHO1* constitute loss-of-function alleles: whereas the mutation in the µHD domain results in loss-of-interaction with its main interacting partners and partial dissociation from the plasma membrane, the mutation in the F-BAR domain alters the subcellular localisation of the FCHO1 protein. Irrespective of their nature, all mutations ultimately result in inefficient nucleation of CCV.

**FCHO1 regulates endocytosis of the T-cell receptor.** The *FCHO1* mutations identified in our patients predominantly result in a severe T-cell defect while other cells of the immune system appear largely unaffected. The key designate of T-cell fate during ontogeny is the quality and strength of the T-cell receptor (TCR) signal[16–18]. The TCR has no intrinsic catalytic activity. TCR-dependent signal propagation requires a multi-protein complex comprises TCR α- and β-chains non-covalently coupled to immunoreceptor tyrosine-based activation motif (ITAM)-rich CD3 ε, γ, δ and ζ molecules. Internalisation of the CD3:TCR complex depends on the formation of CCP[19–24]. We hypothesised that FCHO1 may be involved in TCR internalisation during T-cell activation. To test this, we took advantage of the CD4-positive human Jurkat T-cell lymphoma line in which we deleted the endogenous *FCHO1* gene using CRISPR/Cas9 gene editing. FCHO1-deficient Jurkat clones were reconstituted with either wt or mutated FCHO1 using retroviral vectors encoding a bicistronic FCHO1 and GFP cDNA separated by an internal ribosomal entry site (IRES). In order to understand the effect of FCHO1 on TCR internalisation and clustering, we analysed TCR distribution upon stimulation using confocal microscopy. After 60 min of TCR triggering by an α-CD3 monoclonal Ab, large intracellular CD3-positive puncta were formed in wt cells, whereas in FCHO1 ko clones CD3 molecules remained in diffuse form (Fig. 4a and Supplementary Fig. 8). Knockout clones reconstituted with wt FCHO1 formed large CD3-positive puncta, essentially indistinguishable from those in wt cells. In contrast, none of the FCHO1 mutants were able to rescue the phenotype and thus nearly all CD3 molecules remained diffused upon TCR-activation (Fig. 4a). We also noted plasma membrane invagination and nucleus segmentation in FCHO1-deficient Jurkat cells. This effect was not observed in SK-MEL-2 cells (Figs. 2b–d and 4a, and Supplementary Fig. 5). It is therefore possible, that FCHO1 deficiency has a more general effect on plasma membrane structure in T cells.

To assess TCR internalisation in a quantitative manner, we next measured the intracellular accumulation of CD3:TCR complexes upon anti-CD3-mediated TCR triggering over time using flow cytometry. Consistent with our confocal microscopy studies, we noted that FCHO1 ko cells accumulated approximately two-fold less CD3:TCR complexes when compared to wt cells (Fig. 4b). Finally, as the TCR cross-linking and its subsequent internalisation are essential for quality and strength of the triggered signal, we assessed whether FCHO1 directly

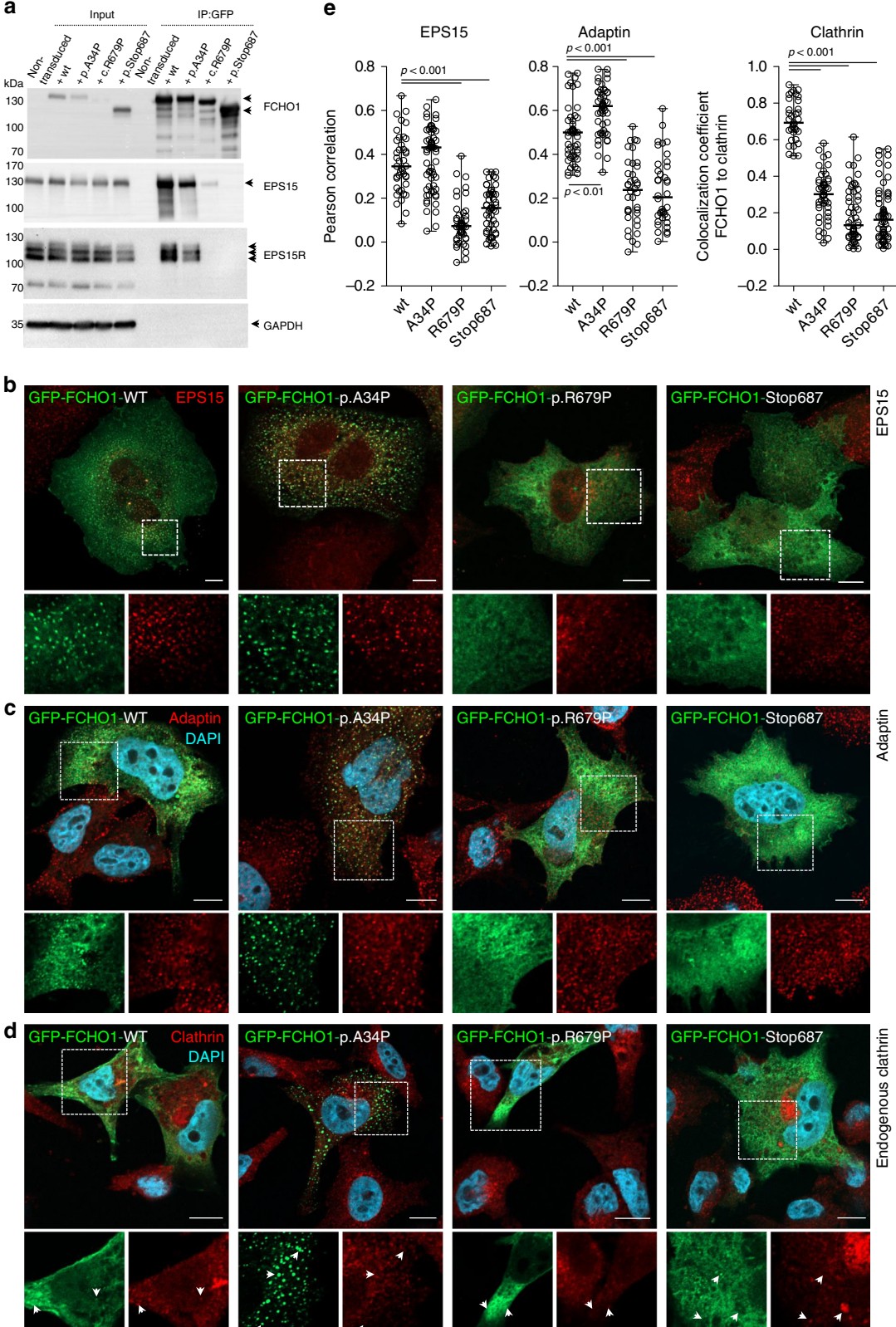

modulated TCR responsiveness. To this end, Jurkat cells sufficient or deficient for FCHO1 were stimulated with an α-CD3 antibody and assessed for release of intracellular $Ca^{2+}$. When compared to controls, FCHO1-deficient cells released less $Ca^{2+}$ upon CD3: TCR triggering (Fig. 4c). Furthermore, only reconstitution with

wt but not mutated forms of FCHO1 restored normal levels of $Ca^{2+}$, directly demonstrating that physiological TCR signalling depends on FCHO1.

During CME adaptor proteins recognise cargoes at the cell surface and direct them to the clathrin pits. FCHO1 binds its

**Fig. 2 Patient-associated mutations alter either binding properties or subcellular localisation of the FCHO1 protein. a** Whole-cell lysates from HEK293T cells overexpressing either wt or indicated mutant GFP-FCHO1 fusion proteins were used for immunoprecipitation. Specific bands are indicated with arrows. Anti-FCHO1 or anti-GFP antibodies were used independently to detect FCHO1-specific bands. Representative data of three independent experiments are shown. Uncropped blots are shown in Supplementary Fig. 4. **b–d** FCHO1-deficient SK-MEL-2 cells expressing RFP-tagged clathrin light chain from endogenous locus (CLTA$^{RFP/wt}$) were transiently transfected with either wt or mutated GFP-FCHO1 and fixed 24 to 36 h post transfection. Representative confocal microscopy pictures show that the F-BAR-domain-associated mutation p.A34P alters the subcellular localisation of FCHO1 and leads to the formation of large aggregates dissociated from the plasma membrane. The µHD domain-associated mutations (p.R679P and p.Stop687) abolish the interaction of FCHO1 with its interacting partners EPS15, and adaptin. All mutations obliterate interaction with endogenous clathrin. Enlarged and colour-separated regions corresponding to boxed areas are shown below each main picture. In **d** arrows indicate presumptive interaction of clathrin with wild-type FCHO1 and lack of such interaction for all tested mutants. Scale bar represents 5 µm for main pictures and 10 µm for enlarged regions. Colour code: **b–d** GFP-FCHO1, green; DAPI, blue, **b** EPS15, **c** adaptin, **d** clathrin–red. **e** Quantification of data shown in **b** to **d**. Pearson correlation or co-localisation coefficients of FCHO1 wild-type and all tested mutants with EPS15, adaptin and clathrin. Pooled data of two to three independent experiments are depicted. Each symbol represents one region of 25 µm². Up to three regions per cells were quantified. Horizontal lines indicate the median, whiskers indicate the range (min to max). Statistical analysis of significance was performed using one-way ANOVA test followed by Tukey's multiple comparison test to assess differences between groups. Source data are provided as a Source Data file.

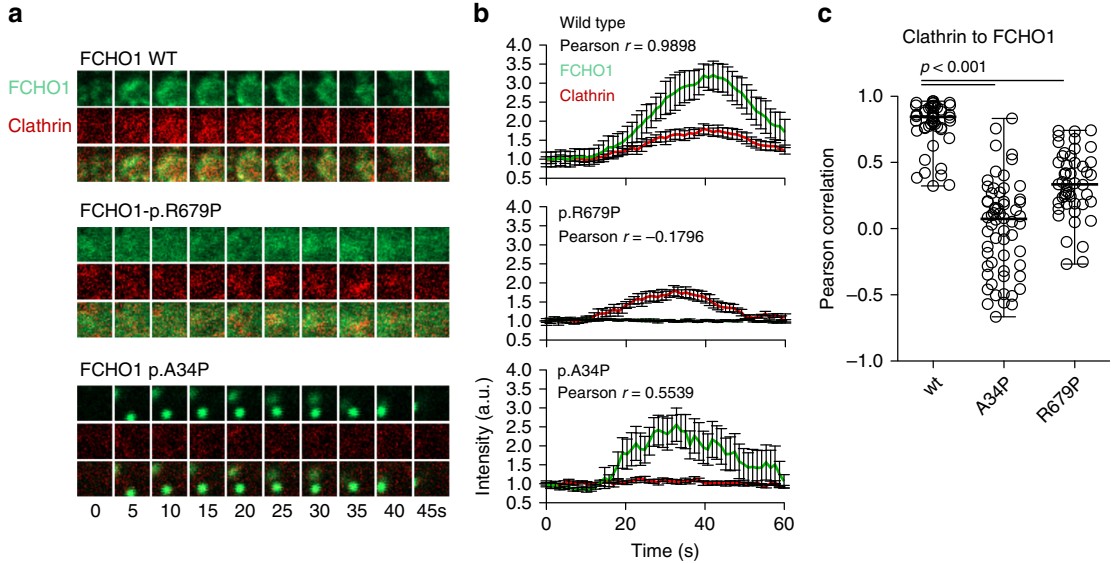

**Fig. 3 Mutations in both µHD and F-BAR domains of FCHO1 prevent nucleation of clathrin-coated pits (CCP). a** The dynamics of FCHO1-medicated nucleation of CCP in FCHO1-deficient SK-MEL-2 cells expressing RFP-tagged clathrin light chain from endogenous locus (CLTA$^{RFP/wt}$) transduced with either wild-type (upper panel), µHD-mutated (p.R679–middle panel) or F-BAR-mutated (p.A34P–bottom panel) GFP-FCHO1 fusion protein. Two micrometre-wide sections of representative movies are shown. Full movies are available in supplemental materials. In the middle panel, contrast was reduced and brightness was increased as to show a diffused signal of GFP at the plasma membrane. **b** Time dependence of the fluorescent intensity of FCHO1 (green) and endogenous clathrin (red) averaged from nine independent movies. Only the fluorescence of wild-type but not mutant FCHO1 correlates with clathrin. Each channel was normalised to the background and the initial fluorescence was set to 1. Error bars represent SEM of mean fluorescence intensity, $n = 9$ biologically independent cells from minimum three independent experiments. **c** Pearson correlation of FCHO1 and clathrin from individual movies. Pooled data from three independent experiments. Each symbol represents one square region of 25 µm². Up to three regions per cells were quantified. Horizontal lines indicate the median. Statistical analysis of significance was performed using one-way ANOVA test followed by Tukey's multiple comparison test to assess differences between groups Source data are provided as a Source Data file.

cargo through its µHD domain[25,26]. It has been proposed that Syp1, the yeast FCHO1 homologue, recognises DxY motifs on its cargo[26]. We hypothesised that human FCHO1 may also recognise its cargo via DxY motifs, in particular since CD3ε and CD3γ have DxY motifs in their cytoplasmic domain (Supplementary Fig. 9a). While we could confirm interaction of FCHO1 and EPS15 in Jurkat cells stably transduced with N'- or C'-flag FCHO1 fusion proteins, we could not observe any direct interaction with either of the CD3 molecules (Supplementary Fig. 9c–f). Thus, although CD3 molecules possess the putative sorting motifs recognised by FCHO1, we could not provide experimental evidence for direct FCHO1-CD3 interaction.

In sum, we provide evidence that FCHO1 plays a role in TCR-dependent T-cell activation. It affects TCR clustering upon

receptor triggering and modulates its internalisation. Finally, FCHO1 deficiency results in impaired mobilisation of Ca$^{2+}$, directly linking the FCHO1 to TCR-associated signalling.

**FCHO1 deficiency does not affect entry of VSV-G pseudotyped HIV-1.** Infection of cells with vesicular stomatitis virus (VSV) is strongly dependent on CME. CME of VSV particles is initiated after interaction of the viral glycoprotein (VSV-G) with the cellular LDL (low-density lipoprotein) receptor[27]. Entry of VSV or VSV-G pseudotyped lentiviruses into the cytoplasm occurs after endosomal acidification, which induces a conformational change of VSV-G followed by fusion of viral and cellular membranes[28].

To evaluate whether the absence of FCHO1 in Jurkat T cells has an impact on CME-dependent virus infection, Jurkat wt and

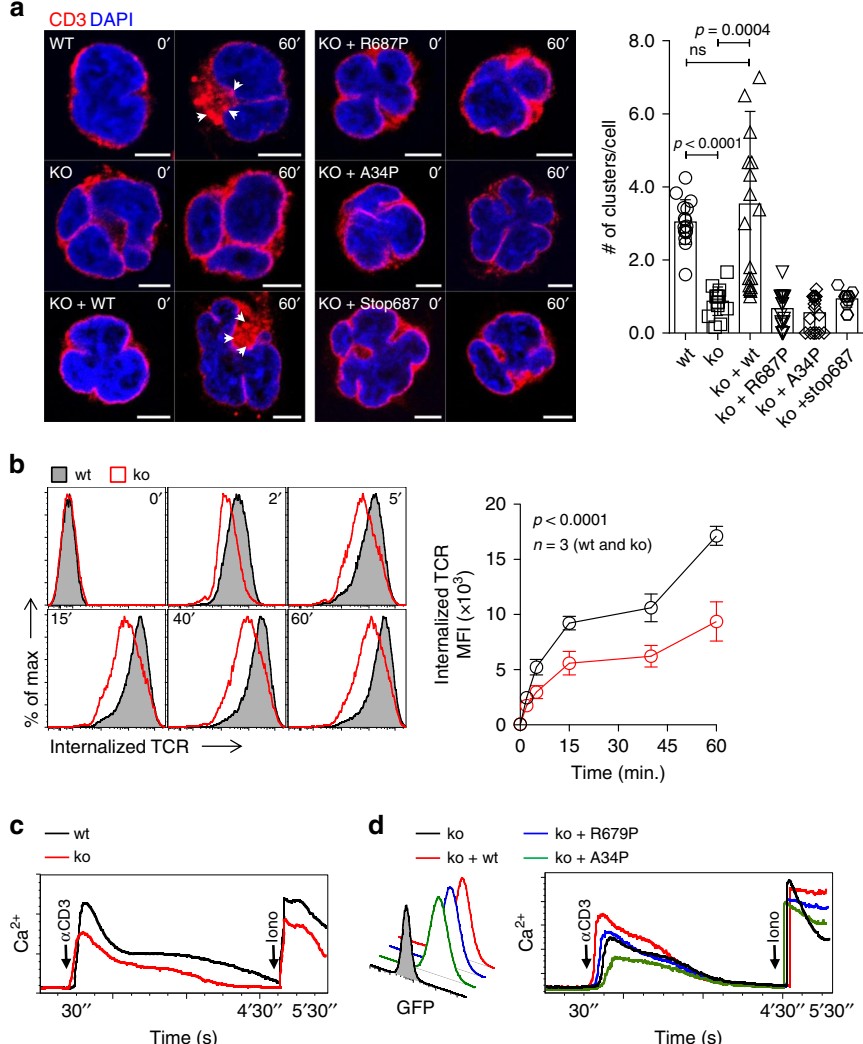

**Fig. 4 FCHO1 deficiency impairs TCR internalisation. a** FCHO1-sufficient and -deficient clones of Jurkat cells were used in the heterologous system, in which ko clones were left either non-transduced or stably transduced with wt or FCHo1 construct carrying one of the patient-associated mutations, as indicated. Cells were mock-treated (0′) or stimulated with anti-CD3 Ab (60′) at 37 °C, fixed and stained for CD3 and DAPI. Representative confocal microscopy pictures show that only wt FCHO1 facilitates the formation of CD3 puncta upon stimulation. Scale bar 5 μm. White arrowheads indicate CD3 puncta. The chart summarises data of three independent experiments in which an average number of CD3 puncta per cell is shown. Each point indicates an average number of puncta per cell that could be found in one field of view. Statistical analysis of significance was performed using ANOVA test followed by Sidak's multiple comparison test to assess differences between groups, whiskers indicate the range (5–95 percentile). **b** FACS analysis of TCR internalisation in wt or FCHO1 ko Jurkat clones. Jurkat cells were stained with anti-CD3 Ab in cold and then TCR internalisation was assessed over time at 37 °C in the presence of anti-mouse F(ab′)$_2$ fragments labelled with Ax647. At indicated time points remaining surface TCRs were stripped, thus fluorescent signal corresponds to the internalised TCR only. The chart summarises data of one representative experiment out of three, $n = 3$ clones per genotype. **c** Intracellular Ca$^{2+}$ flux upon TCR stimulation. Wt or FCHO1 ko Jurkat clones were loaded with Ca$^{2+}$-sensitive FuraRed and Fluo-4 dyes and stimulated with anti-CD3 Ab and then Ca$^{2+}$ flux was recorded flow cytometrically over time. α-CD3 and Iono indicate time points of respective stimulations. Data are representative of three independent experiments in which minimum three different clones of each genotype were analysed. **d** Ca$^{2+}$ flux of FCHO1-deficient clones upon reconstitution with either wt or indicated mutants of FCHO1. GFP histogram indicates transduction efficiency. Intracellular Ca$^{2+}$ flux was assessed as in **c**, representative data of two independent experiments are shown. Two FCHO1-deficient clones were analysed. **b, e** Statistical analysis was performed using two-way ANOVA ($p$-values for the effect of the genotype). Source data are provided as a Source Data file.

FCHO1 ko clones were challenged with HIV-1 particles, that are devoid of their own envelope glycoprotein but had been pseudotyped with VSV-G. The interaction of T cells with VSV-G HIV-1 was monitored using two established readouts, i.e., virion fusion and productive HIV-1 infection. The quantitative assessment of fusion of virions is based on the incorporation of a BlaM-Vpr chimeric fusion protein into VSV-G HIV-1 particles and their subsequent delivery into the cytoplasm of T cells as a result of virion fusion. Cleavage of the fluorescent CCF2 dye,

which is loaded into target cells, allows for detection of fusion events by flow cytometry[29–31]. The successful infection of VSV-G HIV-1 pseudotypes can be quantified by intracellular staining for HIV-1 p24 antigen[32].

Jurkat wt and FCHO1 ko clones were challenged with different multiplicities of infection of either VSV-G HIV-1ΔEnv (BlaM-Vpr) (Fig. 5a and Supplementary Fig. 10a) or VSV-G HIV-1ΔEnv (Fig. 5b and Supplementary Fig. 10b). FCHO1-deficient Jurkat T clones displayed a susceptibility to VSV-G pseudotyped HIV-1,

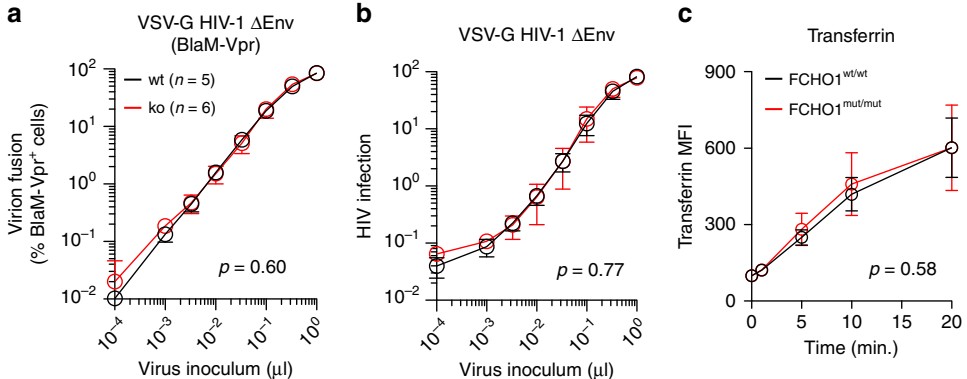

**Fig. 5 FCHO1 deficiency does not alter global clathrin-mediated endocytosis. a, b** Fusion of VSV-G HIV-1ΔEnv (BlaM-Vpr) virions (**a**) or infection by VSV-G HIV-1ΔEnv (**b**) of Jurkat cells. Wt or FCHO1-deficient clones were challenged with increasing volumes of the indicated VSV-G HIV-1ΔEnv. Virion fusion was monitored by flow cytometry as previously reported[29–31] and the percentage of cleaved CCF2+/BlaM-Vpr+ cells is plotted relative to the virus inoculum. Infection of VSV-G HIV-1ΔEnv was monitored by intracellular HIV-1 p24 staining two days post challenge and the relative percentage of p24-positive cells is plotted relative to the virus inoculum[32]. Arithmetic means and standard errors are shown of results obtained for the indicated number of clones from either virion fusion (**e**) or infection (**f**). **c** Patient fibroblast or healthy donor fibroblast were subjected to the transferrin uptake assay. Cells were exposed to fluorescently labelled transferrin for the indicated time at 37 °C and subsequently analysed by FACS. Data are pooled of four independent assays; error bars represent standard deviation. **a–c** Statistical analysis was performed using two-way ANOVA (p-values for the effect of the genotype). Source data are provided as a Source Data file.

monitored by both virion fusion and productive infection, that was indistinguishable from that of wt clones. As controls of specificity, fusion of HIV-1 wt, but not of VSV-G HIV-1 ΔEnv, was inhibited by the peptidic HIV-1 fusion inhibitor T20 (Supplementary Fig. 11a) and, importantly, infection by VSV-G HIV-1 ΔEnv was blocked by the V-ATPase inhibitor bafilomycin A1, which prevents endosome acidification (Supplementary Fig. 11b).

Taken together, FCHO1 deficiency in Jurkat T cells does not functionally impair CME in the context of VSV-G-mediated entry and infection of a lentivirus pseudotype.

**FCHO1 deficiency does not alter global endocytosis.** To further assess whether FCHO1 deficiency has a global impact on CME, we tested transferrin receptor internalisation, a well-recognised model of a clathrin-dependent process[33]. To this end, we incubated patient and healthy donor fibroblasts with fluorescently labelled transferrin in the cold and, following a temperature shift, its internalisation was monitored over time. Both wild-type and FCHO1-deficient primary cells were able to internalise transferrin through the TfR comparably (Fig. 5c), suggesting that FCHO1 deficiency does not affect general CME endocytosis, but selectively CME endocytosis of certain molecules.

**FCHO1 modulates function of primary human T cells.** The paucity of T cells in peripheral blood of most FCHO1-deficient patients precluded an in-depth investigation of primary lymphocytes. However, we were able to test functional consequences of FCHO1 deficiency on T cells isolated from the patient of kindred C. We assessed proliferation of T cells and their capacity to produce cytokines in response to TCR stimulation. First, we determined the frequency of T cells in peripheral blood of patient C1 (Fig. 6a). Although largely reduced in number when compared to her heterozygous siblings, peripheral blood T cells were abundant enough to perform a functional assay. To this end, PBMCs were labelled with CFSE and stimulated with α-CD3 and α-CD28 Abs. Cell proliferation and cytokine production were assessed after 3 and 5 days. At both time points, CD4 and CD8 T cells of healthy siblings responded vigorously to the stimulation whereas patient T cells failed to proliferate (Fig. 6b). Similarly,

FCHO1-deficient T cells produced considerably lower levels of IL-2 and IFN-γ. In contrast, secretion of TNF-α was comparable to healthy control cells, and IL-4 secretion was only marginally dependent of FCHO1 function (Fig. 6c). Of note, T cells from the heterozygous mother (and to a lesser extent the father) produced less cytokines when compared to FCHO1wt/wt control T cells. The reason for this observation remains unclear. While a mild dominant-negative effect cannot be excluded, there is no in vivo evidence of T-cell dysfunction in healthy parents. (Fig. 6c).

In summary, FCHO1 deficiency impairs T-cell development and responsiveness to TCR stimulation.

**Inhibition of CME arrests development of murine T cells in vitro.** Our data highlighted a mechanistic link between impaired CME and ensuing T-cell deficiency. To further validate this notion, we employed OP9-DL1 and OP9 co-culture systems allowing us to study in vitro T-cell differentiation, as well as differentiation of B cells and myeloid cells[34]. Purified thymus-derived, T-cell-committed double-negative (DN) three progenitors were co-cultured with OP9 DL1 cells for 10 days until ~50% of them co-expressed CD4 and CD8 co-receptors (double-positive, DP). At day 5 of culture, we exposed the cells to chlorpromazine, a known inhibitor of CME[35] (Fig. 7a). In a concentration-dependent manner, we observed inhibition of developmental transition between DN3 to DP stage. Whereas > 50% of cells reached the DP stage in the absence of CME inhibition, only 30–35% of chlorpromazine-treated showed progression to the DP stage. Even though minuscule surface expression levels of pre-TCR at this stage of thymocyte development prevented any direct measurements of TCR internalisation, it is known that thymocyte differentiation from DN3 to DP strongly depends on the quality of pre-TCR signal[36,37].

To exclude the general cytotoxic effect of chlorpromazine we co-cultured bone marrow-derived lineage-Sca-1+CD117+ (LSK) cells, the most versatile progenitors, which in similar conditions develop to nearly all hematopoietic lineages (albeit with different kinetics), and assessed how chlorpromazine affects development of various lymphoid and non-lymphoid cells. To this end, LSK co-cultured on OP9-DL1 cells gave rise to all pre-TCR independent thymocyte populations in an indistinguishable

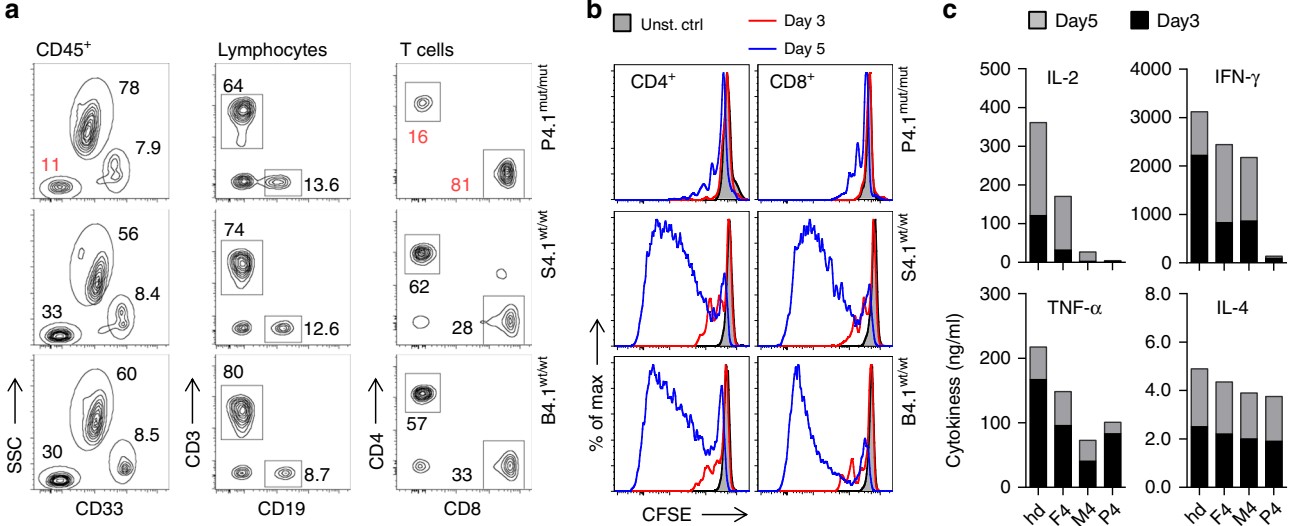

**Fig. 6 Patient-associated mutation in *FCHO1* gene alters development and activation of T lymphocytes. a** Density plots of blood leucocytes (CD45⁺) of the index case (upper row) homozygous for a mutation in the *FCHO1* locus and her siblings carrying the heterozygous mutation (middle and bottom rows). T helper cells were defined as CD45^hiCD33⁻SSC^loCD3⁺CD4⁺ and cytotoxic T cells were CD45^hiCD33⁻SSC^loCD3⁺CD8⁺. Numbers adjacent to the gates indicate percentages. **b** Histograms of CFSE-labelled T lymphocytes stimulated with anti-CD3 and anti-CD28 Ab for 3 (red line) or 5 (blue line) days. Unstimulated controls are depicted in grey. **c** Cytokine production by lymphocytes of patient and family members after anti-CD3 and anti-CD28 stimulation for the indicated periods of time. IL-2, IFN-γ, TNF-α and IL-4 were determined using cytometric bead assays. **a–c** Data are representative of two independent experiments, except data of day 3 shown in **c**, which was assessed once. Source data are provided as a Source Data file.

manner irrespective of presence or absence of chlorpromazine (Fig. 7b). Similarly, presence or absence of chlorpromazine in LSK and OP9 co-culture did not interfere with development of neither B220⁺CD19⁺ B-cell committed progenitors nor CD11b⁺Gr-1⁺ neutrophilic granulocytes (Fig. 7c, d).

Thus, chlorpromazine, a chemical inhibitor of CME, shows rather selective effects on the DN3-DP transition of thymocyte differentiation, known to be particularly vulnerable to disturbances of pre-TCR-signalling strength[36,37]. Taken together, these data indirectly support our concept that T-cell differentiation in FCHO1 deficiency is at least partially dependent on perturbed TCR internalisation/signalling.

## Discussion

Here, we identify autosomal recessive FCHO1 deficiency as a human genetic defect associated with combined immunodeficiency (CID). We show that six different mutations in the *FCHO1* gene, either point mutations resulting in amino acid substitutions, premature stop codons or affecting pre-mRNA splicing are deleterious for FCHO1 function. T-cell deficiency predisposes affected patients to severe and persistent viral and fungal infections. Hypogammaglobulinemia is seen in all patients, except G1. It remains currently unclear whether B-cell defects are intrinsic or strictly dependent on defective T cells.

FCHO1 deficiency predisposes not only to infections but also to lymphoma[13]. Several patients died secondary to malignancies before definitive therapy in form of an allogeneic hematopoietic stem cell transplant could be done. Thus, FCHO1 deficiency warrants rapid genetic diagnosis and provision of access to definitive cure.

Our studies have provided insights into the structural and functional biology of FCHO1. In a structure-guided prediction, Ma and colleagues[14] concluded that a minimum of two aa substitutions in FCHO1 (at positions K877E + R879A), located directly in the peptide-binding groove, are required to abolish interaction of the domain with its main interacting partner EPS15. Here, we show that a single substitution at R679P,

spatially distant from the groove, is sufficient to severely alter this interaction. Although localised opposite of the groove, replacement of the charged side chain of Arg with a nonpolar and rigid ring of Pro is sufficient to alter the domain function. Three mutations resulting in a premature stop codon at the beginning of μHD domain further strengthen their significance for FCHO1 function. Alterations in the F-BAR domain also prove to be deleterious, yet in a different mode. Substitution of Ala at position 34 with Pro, a known alpha-helix breaker, leads to dissociation of the FCHO1 homodimers from the plasma membrane and hence abolishes the function. Although not directly tested, it is plausible that alternative splicing resulting in loss of 51 aa located within the F-BAR domain disrupts the protein structure. From a clinical point of view, the severity of the immunodeficiency, however, cannot directly be correlated to the type of the *FCHO1* mutation.

Given the central role of FCHO1 for the formation of clathrin-coated pits and endocytosis it is surprising that FCHO1 deficiency results in T-cell immunodeficiency rather than more global defects of development. In contrast, *Caenorhabditis elegans* deficient for FCHO show body malformation and uncoordinated locomotion[38]. FCHO interacts with AP-2 and induces a conformational change, leading to AP-2 activation. Mutations in either FCHO or AP-2 result in a strikingly similar phenotype, indicating that the severe defects seen in FCHO-deficient worms are caused by aberrant activation of AP-2. In vertebrates, however, the interplay between AP-2 and FCHO is more complex. Morpholino-mediated knockdown of Fcho1 in *Danio rerio* causes dorsoventral patterning defects and severe malformation at an early developmental stage, whereas transcriptional silencing of Fcho2 is associated with notochord and somite malformations[38]. In zebrafish, AP-2 deficiency results in a more severe, broader and earlier developmental phenotype than combined Fcho1/2 deficiency, suggesting that AP-2 function is, at least in part, Fcho1/2-independent[39]. The idea that the functional relevance of FCHO has changed during evolution is supported by the observation that a) two FCHO paralogs have emerged[40] and b) FCHO may act as receptor-specific adaptors (e.g., BMP-mediated signalling in zebrafish) rather than universally

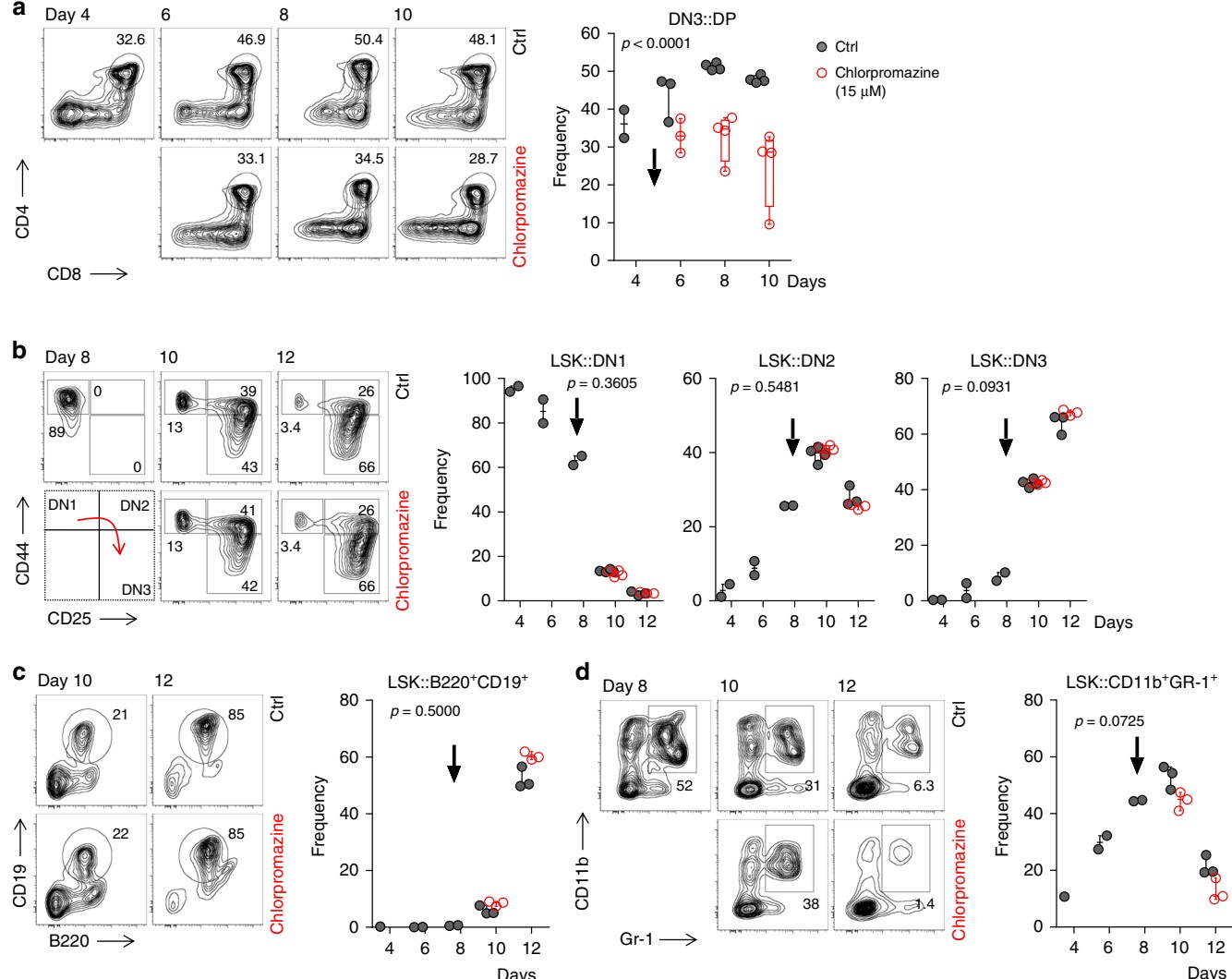

**Fig. 7 Chlorpromazine-induced inhibition of CME arrests development of thymocytes. a** Thymi-derived double-negative three progenitors (DN3: CD4⁻CD8⁻CD44⁺CD25⁻) were sorted and cultured on OP9-DL1 stroma cells in the presence of Flt-3L, SCF and IL-7. At day 5 of co-culture cells were left untreated (ctrl) or treated with chlorpromazine (15 μM) to partially inhibit CME (chlorpromazine IC₅₀ for CME was established at 17.4 μM[35]). Subsequently, development of double-positive (DP: CD4⁺CD8⁺) progenitors was assessed by FACS (left) and quantified (right). **b** Bone marrow-derived LSK progenitors (lineage⁻Sca-1⁺CD117⁺) were sorted and cultured on OP9-DL1 stroma cells in the presence of Flt-3L, SCF and IL-7. Analogous to **a**, at day 8 of culture cells were left untreated (ctrl) or treated with 15 μM chlorpromazine. Effect of partial CME inhibition on early-thymocyte development was assessed by FACS. Double-negative (DN) cells were defined as follows: DN1-CD44⁺CD25⁻; DN2-CD44⁺CD25⁺ and DN3-CD44⁻CD25⁺. **c, d** Bone marrow-derived LSK progenitors (lineage⁻Sca⁺1⁻CD117⁺) were sorted and cultured on OP9 stroma cells in the presence of Flt-3L, SCF and IL-7. At day 8 of culture cells were left untreated (ctrl) or treated with chlorpromazine (15 μM) to partially inhibit CME. Subsequently, development of **c** B-cell committed progenitors defined as B220⁺CD19⁺ and **d** granulocytes (CD11bint⁻hiGr-1⁺) was assessed by FACS (left) and quantified (right). **a–d** Density FACS plots and charts are representative of two independent experiments where one to four wells were measured at each indicated time point. Number adjacent or within the gates indicate frequency. Each dot on the chart represents one well. Whiskers indicate the range (min to max). Arrows indicate time points when chlorpromazine was added. Statistical analysis was performed using two-way ANOVA (p-values for effect of chlorpromazine). Source data are provided as a Source Data file.

important initiators of clathrin-mediated endocytosis[39]. This notion is also supported by the lack of functional consequences of FCHO1 deficiency observed on both VSV-G-mediated virus entry and TfR endocytosis in T cells. Both processes strictly rely on CME[27,33], but are apparently not affected in FCHO1-deficient cells. We conclude that FCHO1 deficiency does not generally affect cellular processes that require CME, underscoring a selective role of FCHO1.

This concept is further corroborated by a series of other studies focusing on central elements orchestrating clathrin-mediated endocytosis. Defects in prime molecules such as clathrin, epsin, AP-2 and dynamin, result in embryonic lethality in model organisms (reviewed in ref. [5]). Mutations in proteins that are deemed to

be less pivotal for the clathrin-mediated endocytosis have been linked to a variety of diseases such as cancer, neuro-psychiatric disorders, metabolic syndromes, but not to inherited defects of the immune system[8]. Of note, a large number of FCHO1 and FCHO2 interacting partners have been identified, yet none of them have been linked to the adaptive immune system[5,25,38,39,41,42].

The quality of signal transmitted by pre-T-cell receptor (pre-TCR) and TCR is essential for T-cell development and homoeostasis. During T-cell ontogeny, there are three stages in which such signal is critically required for selection, differentiation and commitment of developing thymocytes, respectively (reviewed in refs. [43,44]).

The precise localisation of the TCR that is continuously internalised and recycled back to the plasma membrane is of critical importance for the quality of signal transduction. The TCR can be internalised via clathrin-dependent[19–21,45] or clathrin-independent mechanisms[46]. This process may be ligand-dependent or ligand-independent[19,20,22,44,47,48]. Even though previous studies have linked TCR internalisation to a di-leucine sorting motif present in the CD3γδ subunit of the TCR:CD3 complex[49,50], we could not provide definitive evidence that FCHO1 directly binds to the CD3γδ sorting motifs. Nevertheless, we have shown that FCHO1 deficiency results in impaired TCR internalisation. Our data thus suggest that (a) we were not in the position to prove FCHO1/CD3 interaction using crude biochemical methods or (b) that FCHO1 interacts with other membrane-bound molecules to indirectly induce TCR internalisation.

In theory, FCHO1-dependent TCR internalisation can serve several purposes: (1) activation signals could be amplified via the scaffolding on early endosomes, (2) activation signals could be attenuated via lysosomal degradation of the TCR or (3) recirculation of the TCR to immune synapses via recycling endosomes may modulate TCR-dependent activation signals upon encountering antigen presenting cells. Integrating signal strengths relies on spatiotemporal organisation of the TCR and associated signalling molecules. Although the quality of the signal provided by TCR is essential for the outcome of thymopoiesis, our studies in FCHO1 deficiency do not provide definitive answers on detailed mechanisms of FCHO1 in T-cell differentiation. Indirectly, the link between CME and TCR-signalling is highlighted by our in vitro T-cell differentiation studies. In the presence of chlorpromazine, a chemical inhibitor of CME, we observed a rather specific effect of differentiation blockade at the DN3-DP transition. This step is known to be highly dependent on TCR-signalling strength[36,37].

Very recently, Calzoni et al.[51] published a short letter and reported biallelic mutations in FCHO1 in four families. The phenotypes of these patients resembled the phenotype of our patients, but no functional experiments or proof-of-causality was provided. Based on experiments in activated T-cell blasts, the authors concluded that in the absence of FCHO1, CME is globally affected. In contrast, our data support the concept that FCHO1 does not globally affect CME.

In sum, our studies unravel a previously unrecognised role of FCHO1 in orchestrating the T-cell development and function. Our discovery also exemplifies how systematic studies of patients with inherited disorders help to uncover genes and pathways that were previously not associated with the function of the immune system. FCHO1 deficiency thus highlights a critical role of clathrin-mediated endocytosis for the development and function of human T cells.

## Methods

**Patients**. Patients were referred to the clinical and scientific team of Professor Christoph Klein for further investigations. Informed consent/assent for the genetic and immunological studies, as well as their publication was obtained from all legal representatives and patients. Genetic and functional studies on biosamples from patients and their relatives were performed under the framework of a scientific project entitled "Genetic characterisation of congenital bone marrow failure and immunodeficiency syndromes". This study was approved in 2011 by the ethics committee at LMU (438-11) and includes permission to publish the results.

**Whole-exome sequencing and variant filtering**. For family A we performed whole-exome sequencing (WES) for patient and father using Agilent V4 + UTR library preparation and a SOLiD sequencing platform. WES for the patient from the family B, the patient from family C, as well as for patient, mother and siblings (from family D) was performed after Agilent library preparation (V5 + UTR or V6 + UTR (SY265)) using an IlluminaNextSeq 500 platform. BWA (version 0.7.15) was used to align short reads to the human reference genome Grch37.p13. Variants were called and recalibrated according to the best practice pipeline by GATK (version 3.6). The final variants were then annotated with VEP release 85. A custom in-house database was used to filter variants to be rare (not reported in gnomAD or ExAC), as well as severe effects according to the Ensembl guidelines.

Effects of filtered variants on protein were predicted with SIFT[52] and PolyPhen-2[53]. The remaining variants were compiled and filtered for rare homozygous and compound heterozygous mutations following a pattern of autosomal recessive inheritance. Whole-exome sequencing of genomic DNA of kindred F patients was conducted using Illumina sequencing platforms. Bioinformatics analysis for detection of rare sequence variants following Mendelian inheritance patterns were performed as described previously[54].

**Sanger sequencing**. FCHO1 Sanger sequencing was performed to confirm WES-detected variants and their segregation with the clinical phenotype across the family members. Genomic DNA was PCR-amplified using OneTaq Polymerase (NEB), specific primers are provided in Supplementary Table 3. Amplicons were sequenced either in-house or by using the commercial service of Eurofins Genomics.

**Structural analysis of FCHO1 mutants**. Crystal structures of FCHO1 domain were modelled using PyMol software with the mutalyzer wizard[55]. Mu homology domain of *Danio rerio* with bound Eps15 peptide (5JP2; https://doi.org/10.2210/pdb5JP2/pdb) and human FCHO2 F-Bar domain (2v0O; https://doi.org/10.2210/pdb2V0O/pdb) were chosen as template for structure modelling. To calculate effect of point mutations on the structures, rotamer configurations of the highest probability were chosen.

**Cell lines**. The SK-MEL-2 cell line, engineered using zinc finger nucleases (ZNF) genome editing to stably express RFP under the endogenous human clathrin light chain A locus (CLTA-RFP), was kindly provided by David G. Drubin, University of Berkeley[15]. The cell line with only one RFP-tagged CLTA locus was used to minimise putative side effects. Cells were maintained in Dulbecco's modified Eagle medium (DMEM)/F12 medium (ThermoFisher Scientific) supplemented with 10% fetal calf serum (FCS) (ThermoFisher Scientific), 100 U/ml of both penicillin and streptomycin (Gibco). Jurkat cells were purchased from ATCC, USA. They were maintained in RPMI1640 medium (ThermoFisher Scientific) supplemented with 10% FCS (ThermoFisher Scientific), 100 U/ml of both penicillin and streptomycin (Gibco) and 2 mM glutamine (Gibco). In all, 100 U/ml of penicillin and streptomycin (Gibco) HEK293T cells and NIH-3T3 cells (both from DSMZ–German Collection of Microorganisms and Cell Cultures) were maintained in DMEM (ThermoFisher Scientific) supplemented with 10% FCS (ThermoFisher Scientific), 100 U/ml each of penicillin and streptomycin (Gibco). Patient fibroblasts were grown in Iscove's Modified Dulbecco's Medium supplemented with 10% FCS. EBV-LCL were maintained in RPMI with 10% FCS. All cell lines were routinely tested for mycoplasma and were mycoplasma-negative throughout the study.

**Plasmid DNA cloning**. Full-length cDNA of the human FCHO1 isoform b (IMAGE 5757146), cloned into the pEGFP-C3 vector with modified MCS (EcoRI/SalI) was kindly provided by Emmanuel Boucrot (UCL London) and Harvey McMahon (MRC-LMB, Cambridge, UK)[5]. Patient mutations were introduced by site-directed PCR mutagenesis using Q5 site-directed mutagenesis kit (NEB) according to manufacturer's instruction with specific primers designed with NEBase Changer (NEB). Wt and mutated FCHO1 cDNAs were cloned as either GFP fusion proteins or as IRES-containing bicistronic lentiviral pRRL vectors with or without GFP as a reporter gene, respectively. Full-length cDNA of human clathrin light chain tagged with mRFP was kindly provided by Klemens Rottner from Technical University in Braunschweig[56]. It was cloned to lentiviral pRRL vector for production of virus particles. The correctness of the sequences was routinely monitored by Sanger sequencing.

**CRISPR/Cas9 genome editing**. The genomic locus of FCHO1 (transcript ENST00000594202.1) was designated for gene disruption by inducing double-strand breaks in exon 7 (T1: 5′-GGACGTTCTCCGCTACGGCG AGG-3′) and intron 7 (T2: 5′-GTGTCGTGGGCGCCGCCCAG CGG-3′). Genome editing of SK-MEL-2 and Jurkat cells was done cells using the Alt-R CRISPR-Cas9 technology (IDT technology, Belgium). Upon coincubation of crRNA and ATTOTM 550 (ATTO-TEC, Germany) fluorescent dye-labelled tracRNA, the RNA duplexes were electroporated into target cells along with Cas9 nuclease (SG Cell Line 4D-Nucleofector X Kit and 4D-NucleofectorTM System Lonza, Switzerland). Red fluorescent protein (RFP)-positive cells were single-sorted using a BD FACSAria flow sorter (BD Bioscience, USA).

As endogenous expression of FCHO1 is too low for faithful assessment by western blot, deletion of FCHO1 was confirmed by PCR. The following primer pair was used for validation: (F: 5′-GTGAGCCTGATGAACCCTGGGTGTG-3′, R: 5′-TGCATGTGGGTGACAGAGTGAGAC-3′). Cells carrying homozygous mutations are expected to have a long deletion, spanning between exon 7 and intron 7. A minimum of five clones carrying homozygous mutations resulting in frameshift were used for subsequent experiments (Supplementary Fig. 12). Unmodified clones were used as a FCHO1-positive control. To ensure deletion specificity, the ten most probable off-target sites were tested using direct Sanger sequencing, showing no signs of unspecific cuts.

**Transient transfection**. FCHO1 ko SK-MEL-2 cells and FCHO1 ko Jurkat cells were transfected using calcium phosphate transfection kit (Sigma, USA) and 0.1–1 μg of the various lentiviral plasmids carrying either wt or mutant FCHO1 cDNAs. Cells were typically incubated 24 h to express the constructs before imaging.

**Lentiviral vector particles production and cell transduction**. For production of vesicular stomatitis virus G glycoprotein (VSV-g)-pseudotyped lentiviral particles HEK293T cells were transfected using calcium phosphate transfection kit (Sigma, USA) and 5 μg respective lentiviral vector, 12 μg pcDNA3.GP.4xCTE (which expresses HIV-1 gag- pol), 5 μg pRSV-Rev and 1.5 μg pMD.G (which encodes VSV-g) in the presence of 25 μM chloroquine (Sigma, USA). Eight hours after transfection, medium was exchanged and supernatant containing lentiviral particles was collected after 24, 48 and 72 h post transfection. Viral titre were determined on 3T3 cells.

FCHO1 ko SK-MEL-2 cells and FCHO1 ko Jurkat cells were transduced with lentiviral particles through centrifugation at $900 \times g$ for 4 h at 32 °C in the presence of polybrene (8 μg/ml) (Sigma, USA).

**Production of HIV-1 stocks**. The HIV-1$_{NL4-3}$wt plasmid was obtained from Nathaniel Landau (Alexandria Centre for Life Science, NYU, USA), the HIV-1$_{NL4-3}$ΔEnv plasmid was a kind gift of Oliver T. Fackler (Universitätsklinikum Heidelberg, Germany) and the BlaM-Vpr plasmid was a gift from Thomas J. Hope (Northwestern University, Chicago, USA). HIV-1$_{NL4-3}$ΔEnv and HIV-1$_{NL4-3}$ΔEnv (BlaM-Vpr), both VSV-G pseudotyped, or HIV-1$_{NL4-3}$wt (BlaM-Vpr) stocks were produced by PEI co-transfection of HEK293T cells. Forty-eight hours later, supernatants were collected and filtered through a 0.45 μm Stericup (Millipore). After sucrose cushion (25% in 1x phosphate-buffered saline (PBS_) purification at 24,000 rpm at 4 °C for 1.5 h (Sorvall WX + Ultra series; rotor: SW32, Beckmann Coulter), virus pellets were resuspended in PBS and stored at −80 °C until use.

**Immunoprecipitation and western blot**. To test for protein–protein interactions, FCHO1-deficient cell lines were used. SK-MEL-2 cells overexpressing different variants of GFP-FCHO1 fusion proteins were starved in serum-free medium for 1 h at 37 °C prior the assay. Jurkat lines were starved for 1 h and subsequently stimulated with an α-CD3 antibody (OKT3, 1 mg/ml, BD Biosciences) cross-linked by a goat α-mouse polyclonal antibody (0.5 mg/ml, Jackson ImmunoResearch) for 2 to 20 min at 37 °C. Stimulation was terminated by addition of ice-cold PBS. Cell pellets were lysed in RIPA buffer containing phenylmethylsulfonyl fluoride protease inhibitors for 30 min in cold. Next, the supernatants were collected for further analysis. GFP-tagged FCHO1 proteins were pulled-down by GFP-Trap magnetic beads (ChromoTek GmbH) and N' or C' Flag-tagged FCHo1 by anti-Flag M2 Affinity Gel (Sigma). After incubation, beads were collected and washed two times in RIPA buffer and the pellet was boiled in sample buffer containing SDS for 10 min at 95 °C. Equal amounts of protein were separated by SDS polyacrylamide gel electrophoresis and blotted onto polyvinylidene difluoride (PVDF) membranes using the Trans-Blot Turbo Transfer System (Bio-Rad). Membranes were blocked for 1 h at room temperature in 5% non-fat milk before staining. Following primary antibodies were used: FCHO1–rabbit, polyclonal, PA5-31603, lot Q12081994A, Thermo Scientific or rabbit, polyclonal, 84740, lot GR214150—4, Abcam; EPS15–rabbit, clone D3K8R, Cell Signalling; EPS15R–rabbit, clone EP1146Y, Abcam; adaptin–mouse, clone AP6, Abcam; CD3epsilon–rat, clone OKT3, ThermoFisher Scientific; CD3delta–rabbit, polyclonal, ThermoFisher Scientific; CD3 gamma–rabbit, polyclonal, ThermoFisher Scientific; GAPDH–mouse, clone 6C5, Santa Cruz. All antibodies used are summarised in Supplementary Table 4.

After washing in PBS-T, the PVDF membranes were exposed to horseradish peroxidase-conjugated secondary anti-mouse (BD), anti-rat (CS), or anti-rabbit (CST) Ig antibodies for 1 h at room temperature (RT). Western blots were detected using a chemiluminescent substrate (Pierce Technology) and images were captured on a Chemidoc XRS Imaging System (Bio-Rad Laboratories). Blots were stripped between exposures to different antibodies using a Restore Western Blot Stripping Buffer (Thermo Scientific). Data analysis was performed using Quantity One or Image Lab software (Bio-Rad Laboratories). Uncropped immunoblots are shown in Supplementary Figures.

**Confocal microscopy analysis of fixed samples**. To minimise dominant-negative effects on CCP dynamics resulting from prolonged overexpression of FCHO1 protein, only FCHO1-deficient SK-MEL-2 cell line transiently transduced with N-terminally tagged versions of GFP-FCHO1 were used for experiments. Cells with a low expression of FCHO1 were analysed typically 16 to 20 h post transfection. Cells were plated on glass coverslips (Karl Hecht, 0.13–0.16 mm thickness, diameter 20 mm) in 24-well tissue culture plates and cultured in complete DMEM/F12 medium for min 12 h to facilitate attachment. Coverslips were rinsed with PBS and fixed with 3% formaldehyde (Electron Microscopy Sciences) for 10 min at RT. Next, cells were washed in PBS and autofluorescence was quenched for 15 min using 50 mM NH$_4$Cl at RT. Coverslips were washed once in PBS and incubated for 30 min at RT in PBS containing 0.1% (w/v) bovine serum albumin (BSA; Sigma Aldrich), 0.05% (w/v) saponin (Sigma Aldrich). Further blocking solution was removed and cells were stained with the following primary antibodies: rabbit-α-Eps15 (clone D3K8R, Cell Signalling) or mouse-α-Adaptin (AP6, Abcam) at 4 ºC, overnight.

Subsequently, cells were washed in PBS containing 0.05% (w/v) saponin (Sigma Aldrich), three times for 5 min each. The following secondary antibodies coupled to fluorochromes were used: goat-α-rabbit IgG, AlexaFluor 405 (Invitrogen) and goat-α-mouse IgG, AlexaFluor 633 (Invitrogen).

WGA (wheat germ agglutinin) staining was performed on transfected cells fixed with 3% formaldehyde (Electron Microscopy Sciences) for 10 min at RT. After extensive washing with phenol red free 1xHBSS, WGA conjugated to AlexaFluor 633 (Invitrogen) was added to the cells at concentration 1.25 μg/ml (in phenol red free 1xHBSS) and incubated for 5 min.

Some samples were co-stained with 300 nM DAPI (Invitrogen) for 2 min at RT. Cells were mounted on glass slides using Fluoromount-G (SouthernBiotech) and dried at RT in darkness for minimum 12 h before imaging. Samples were analysed by confocal fluorescent microscopy using the Zeiss LSM880 and Zeiss LSM800 inverted microscopes. Images were collected using 63×/1.4 NA or 40×/1.4 NA oil objectives (Zeiss). Four solid-state 5 and 10 mW laser (405, 488, 561, 640 nm) were used as light source; scanner frequency was 400 Hz; line-averaging 2. All images were obtained with GaAsP high-sensitivity detectors. Pearson correlation or co-localisation coefficient was assessed on 16-bits raw digital files on Zen Blue software (Zeiss). Representative cells from two to three independent experiments were chosen for analysis. Two to four 25 μm$^2$ square regions were selected and minimum ten cells per condition were analysed. To avoid bias during analysis, only one fluorescent channel was active while selecting regions.

**Live-cell imaging**. For live-cell imaging experiments stably transduced SK-MEL-2 cell lines were used. Cells were seeded on μ-dish 35 mm, high Glass Bottom (IBIDI, glass coverslip no.1.5 H, selected quality, 170 μm + /− 5 μm) in complete DMEM/F1224h before imaging. Medium was exchanged directly before start of the experiment. During the experiment dishes were placed into a temperature-controlled chamber on the microscope stage with 95% air, 5% CO$_2$ and 100% humidity. Live-cell imaging data were acquired using a fully motorised inverted confocal microscope (Zeiss LSM800) using either 40×/1.4 NA or 63 ×/1.4 NA oil objectives (Zeiss) under control of Definite Focus for Axio Observer Z1 (Zeiss). Sixteen-bits digital images were obtained with GaAsP high-sensitivity detectors, confocal module and 488 and 561 nm laser lines and a dual (525/50; 605/70) BP filters. Cell regions were selected to allow for 500–950 ms-lasting intervals. Time dependence of the fluorescent intensity of FCHo1 and endogenous clathrin were assessed on movies from three independent experiments, using Zen Blue software (Zeiss). Up to three 25 μm$^2$ regions per cells were quantified. Both GFP and RFP channels were normalised to the background and initial fluorescence was set to 1.

**Fusion of VSV-G pseudotyped HIV-1 ΔEnv to Jurkat T cells**. The virion fusion assay was performed in principle as reported[29] employing HIV-1 particles carrying the HIV-1 Vpr protein fused to β-lactamase (BlaM-Vpr). In brief, after fusion of virus particles to target cells, the incorporated BlaM-Vpr protein is released into the cytoplasm and is able to cleave the CCF2 dye. This leads to a shift of the dye's emission maximum from 520 to 447 nm, which can be detected and quantified by flow cytometry. Jurkat T cells were plated at a density of $2 \times 10^5$ cells per well (96-well conical plate, Corning, New York, USA). Where indicated, the HIV-1 fusion inhibitor T20 (50 μM, enfuvirtide, Roche, Rotkreuz, Switzerland) was added 1 h prior to virus challenge. Serial dilutions of VSV-G HIV-1$_{NL4-3}$ΔEnv (BlaM-Vpr) were performed in PBS. 4 h following challenge with either VSV-G HIV-1$_{NL4-3}$ ΔEnv (BlaM-Vpr) or HIV-1$_{NL4-3}$ (BlaM-Vpr), Jurkat T cells were washed and incubated with the CCF2 dye as previously reported[30,31]. The following day, cells were fixed for 90 min with 4% PFA/PBS and analysed by flow cytometry.

**Infection of Jurkat T cells with VSV-G HIV-1 ΔEnv**. Jurkat T cells were plated at a density of $2 \times 10^5$ cells per well. T20 (50 μM), the V-ATPase inhibitor bafilomycin A1 (100 nM, Sigma Aldrich, St. Louis, USA) or PBS were added 1 h prior to virus challenge. Cells were challenged with VSV-G HIV-1$_{NL4-3}$ΔEnv and 4 h later, 200 μl of fresh culture medium were added. 48 h later, cells were fixed for 90 min in 4% PFA/PBS and HIV infection was monitored using an intracellular p24 staining (anti-p24 antibody, clone KC57-FITC, Beckmann Coulter, Brea, USA) in principle as reported[32].

**Transferrin uptake**. Fibroblasts from healthy donor (HD) and patient (kindred E) were detached using PBS containing 10 mM EDTA and serum-starved for 30 min in DMEM at 37 °C. Cells and transferrin conjugated to AlexaFluor 633 (ThermoFisher Scientific) were washed in glucose buffer (PBS supplemented with 20 mM Glucose and 1% BSA) and cooled down on ice. 25 μg/ml AlexaFluor 633-conjugated transferrin were added to cells and incubated on ice for 10 min. Cells were then transferred to 37 °C for indicated periods of time. Upon indicated time, surface-bound transferrin was stripped by acid wash (PBS supplemented with 0.1 M glycine and 150 mM NaCl at pH 3) and uptake of fluorescent transferrin was determined by flow cytometry. Data are pooled from four independent experiments. Error bars indicate mean ± SD. Statistical analysis using two-way ANOVA followed by Sidak's multiple comparisons test revealed no significant difference in transferrin uptake between HD and patient fibroblasts.

**Flow cytometry**. Blood samples were washed with PBS and stained with the following antibodies for 20 min at RT: α-CD45 BV711 or APC (HI30), α-CD33 PE-Cy7 (P67.6), α-CD3 PE (HIT3a), α-CD19 FITC (HIB19), α-CD8α APC (RPA-T8), α-CD4 PE-Cy7 (A161A1) all from Biolegend. All antibodies used are summarised in Supplementary Table 4. Red blood cells were lysed using 1× BD FACS Lysing Solution (BD Biosciences) according to the manufacturer's instructions. The samples were acquired using a LSRFortessa (BD Bioscience) cytometer. Data were analysed using FlowJo Software (TreeStar), v9 and v10. Gating strategy to assess frequency of blood leucocytes is shown in Supplementary Fig. 13. Virion fusion and HIV infection of Jurkat T cells were recorded on a BD FACSLyric (BD Biosciences, Franklin Lakes, USA).

**In vitro stimulation of PBMCs**. PBMC were isolated from blood samples by Ficoll-Hypaque (Pharmacia) density gradient centrifugation. Cells were labeled with 1 μM CFSE (eBioscience) according to the manufacturer's protocol. They were resuspended in complete RPMI1640, plated on 96-well flat bottom plates and stimulated with α-CD3-coupled beads (bio-α-CD3, clone OKT3 (eBioscience) coupled with α-biotin MACSiBeads (MiltenyiBiotec)) at a ratio 10:1, in the presence of 1 mg/ml of soluble α-CD28, clone CD28.2 (eBioscience). Proliferative response was measured after 3 and 5 days. Gating strategy to assess T-cell proliferation is shown in Supplementary Fig. 14. Supernatants (four technical replicates) were tested for the presence of IL-2, IL-4, IFN-γ and TNF-α using human FlowCytomix beads (eBioscience) according to the manufacturer's instructions.

**Assessment of intracellular calcium flux**. Up to five different FCHO1-sufficient or -deficient Jurkat clones were incubated for 1 h in $Ca^{2+}$- and $Mg^{2+}$-free Dulbecco's serum-free medium (Invitrogen) at room temperature at a density of $10^7$ cells/ml. Cells were then loaded with $Ca^{2+}$-sensitive dyes, either Indo-1 or Fluo-4 (3 μM) and FuraRed (6 μM) for 45 min at 37 °C. Further, cells were rested for 30–45 min at 37 °C. After establishing a baseline for 30 s, cells were stimulated with α-CD3 antibody (OKT3, 1 mg/ml, BD Biosciences) and goat-α-mouse polyclonal antibody (0.5 mg/ml, Jackson ImmunoResearch) to allow cross-linking and data acquisition was continued for four additional minutes. To ensure cell viability, 1 min before the end of acquisition, 2 μg/ml ionomycin (Sigma) was added as positive $Ca^{2+}$-flux control. Gating strategy to assess $Ca^{2+}$ release is shown in Supplementary Fig. 15.

**TCR internalisation assays on Jurkat cell lines**. Confocal microscopy: FCHO1-sufficient or -deficient, or stably transduced (with wt or mutant FCHO1 viruses) Jurkat $FCHO1^{-/-}$ cell lines were used to visualise TCR internalisation. Cells were plated on poly-D-lysine (0.1 mg/ml, Sigma Aldrich) and α-CD3 (clone OKT3, 0.5 mg/ml, eBioscience) coated glass coverslips (Karl Hecht, 0.13-0.16 mm thickness, diameter 20 mm) in 24-well tissue culture plates and incubated for indicated time points. Coverslips were rinsed with PBS and cells were fixed in 3% formaldehyde (Electron Microscopy Sciences) for 10 min at RT. Next, cells were washed in PBS and autofluorescence was quenched for 15 min at RT using 50 mM $NH_4Cl$. Coverslips were washed once in PBS and incubated for 30 min at RT in PBS containing 0.1% (w/v) BSA (Sigma Aldrich), 0.05% (w/v) saponin (Sigma Aldrich). Further, blocking solution was removed and cells were stained with mouse-α-CD3 (OKT3, eBioscience) and goat-α-mouse IgG, AlexaFluor 633 (Invitrogen). Samples were co-stained with 300 nM DAPI (Invitrogen) for 2 min at RT. Cells were mounted on glass slides using Fluoromount-G (SouthernBiotech) and dried at RT in darkness for at least 12 h before imaging. Samples were analysed by confocal fluorescent microscopy using the ZEISS LSM800 inverted microscopes (ZEISS) as described above. We used Fiji software[57] to quantify the number of CD3 puncta/cell. The number of puncta/cell was averaged on several random fields of view.

Flow cytometry: FCHO1-sufficient or -deficient Jurkat cells were stained with α-CD3 Ab (OKT3) in cold and TCR internalisation was assessed over time at 37 °C in the presence of anti-mouse F(ab')2 fragments labelled with Ax647. After 2, 5, 15, 45 and 60 min of stimulation the remaining surface TCRs were stripped, thus fluorescent signal corresponds only to the internalised TCR. Gating strategy to assess TCR internalisation is shown in Supplementary Fig. 16.

**Statistics**. Statistical analysis was performed using GraphPadPrism software v.6. Pearson correlation and colocalization coefficient on selected cell fragments were assessed on raw files using Zen Blue software (Zeiss). Cell regions or entire cells are referred to as *n* unless indicated otherwise. No method of randomisation was used, and no samples were excluded from analysis. To avoid bias during analysis, cell regions were selected based on the signal from only one fluorescent channel. No statistical method was used to predetermine sample size for analyses. Two-way analysis of variance (ANOVA) or ANOVA analysis followed by Sidak's multiple comparison test were used to assess differences between groups. $p$-values < 0.05 were considered to be statistically significant.

**Reporting summary**. Further information on research design is available in the Nature Research Reporting Summary linked to this article.

## Data availability

The source data underlying Figs. 2e, 3b, 3c, 4b, 5a–c, 6, 7a–d, Supplementary Figs. 1, 5, 10, 11 are provided as a Source Data file. All other data are available from the corresponding authors upon reasonable request. The identified *FCHO1* mutations have been submitted to the ClinVar database with accession numbers SCV001146883, SCV001146884, SCV001146885, SCV001146886, SCV001146887 and SCV00114688. According to current regulatory frameworks, exome sequencing data cannot be made publicly available. For any further questions, the corresponding authors will share additional data in accordance with regulatory guidelines.

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

## Acknowledgements

We thank the patients and their families for participating in this research. We would like to thank Alper Özcan, Murat Cansever, Fulya Bektas, Ebru Yilmaz, (Erciyes University, Kayseri, Turkey), Can Acıpayam (Kahramanmaraş Sütçü imam University, Kahramanmaraş) and all medical personnel for excellent patient care. We thank Andreas Krueger and Ludger Klein for critical review of this manuscript; Emmanuel Boucrot (University College London, UK) and Harvey McMahon (MRC-LMB, Cambridge, UK) for sharing various constructs of wt FCHO proteins. The SK-MEL-2 cell line with RFP-tagged clathrin light chain was kindly provided by David G. Drubin (UC Berkeley, USA). We are grateful to Jan E. Heil, Monica Stich and Alexander Liebstein from ZEISS laboratories for access to their facilities and technical help on microscopy experiments. The studies were supported by the Bundesministerium für Bildung und Forschung (BMBF) (German PID-NET, grants to K.S., M.H., J.R., and C.K.), the DAAD (Rare disorders of the immune system), the Else-Kröner-Fresenius Stiftung, DFG (Gottfried Wilhelm Leibniz programme and SFB914 (to C.K.), German Research Foundation (DFG LY150/1-1) (to M.Ł.), grants KE742/5-1 and KE742/7-1 (to O.T.K.)) and the Care-for-Rare Foundation.

## Author contributions

M.Ł. and N.Z. initiated the project, designed the studies, performed experiments and analysed the data (pedigrees A-D); L.F. performed IP and microscopy experiments, analysed the data (pedigrees A-D); U.P. designed and performed experiments, analysed the data (pedigree F); M.S performed HIV infection experiments and analysed the data; Y.L. designed and performed CRISPR/Cas9 knockout experiments; Y.F. performed IP experiments (pedigrees A-D); J.P. and S.H. performed bioinformatical analysis of WES data (pedigrees A-D); I.S. performed genotyping and WB experiments, analysed the data (pedigree E); M.R. performed Sanger sequencing and in silico prediction analyses of mutations (pedigrees A-D); K.-W.S. provided clinical data (pedigree A); O.T.K designed and analysed HIV infection experiments; E.Ü provided patient material, clinical data (pedigree C); M.K., T.P. provided patient material, clinical data (pedigree B, D); A.L., A.S. and R.S. provided patient material, clinical data and genetic analysis (pedigree E); M.H. J.R. and K.S. designed the studies, provided patient material, clinical data and genetic analysis (pedigree F and G); C.Ke. and E.K. performed family analyses, RNA and protein detection (Pedigree F). C.K. initiated the project, designed the studies and provided clinical and laboratory resources; (pedigrees A-E); M.Ł., N.Z. and C.K. wrote the manuscript.

## Competing interests

The authors declare no competing interests.
