## [Peer Review File · Nature Communications]

Reviewers' Comments:

Reviewer #1:

Remarks to the Author:

This is an interesting manuscript showing the requirement of the clathrin-mediated endocytosis effector FCH domain only 1 for T cell receptor (TCR) internalisation. The strong aspect of the study is the link between FCHO1 mutations and lymphopenia in human. The main weakness of the work is the fact that these mutations, which totally prevent FCHO1 to associate/initiate with clathrin-coated pits, have an effect only on TCR endocytosis (as mentioned in the discussion lines 292-295 and 306-308). It is a source of concern because the data presented do not allow figuring out if the expression of the mutant prevents clathrin-mediated endocytosis (CME) at all, or just the CME of the few proteins (including TCR) that require FCHO1 to get into clathrin coated pits. The authors even touch that issue, l.219-220, suggesting that the absence of FCHO1 may impact the whole organisation of T cell plasma membrane. Hence, the defect in T cells activation could be due to a disorganisation of the plasma membrane, and not to a defect in TCR internalisation.

For these reasons, the authors ABSOLUTELY have to provide an example of a receptor, known to be internalised through CME, which is not affected by the absence of FCHO1. It would also have more impact if these cells come from the patients. In the same line, the authors have to perform an experiment to show that the general organisation of the plasma membrane is not affected in T cells (Jurkats) that lack FCHO1. For instance using FCS or image correlation. Or at least, to show that there no difference in the formation of TCR microclusters on antibody-coated glass (the Samelson way) or on supported bilayers (the Dustin way).

In addition, many relevant control experiments are missing (see specific comments).

More specific comments:

Effects of patient-associated mutations on FCHO1 function

- 1) There is some confusion in figure 2. The cells are supposed to express Clathrin-RFP (l. 166). Yet, the red channel shows EPS15 in b, adaptin in c and indeed clathrin in d. Legends and text have to be amended.
- 2) l. 171-172, the images do not really allow to see if the punctate structures in the F-BAR domain mutant are dissociated from the plasma membrane. A xz or yz image of those cells is required to support this statement.
- 3) Figure 2 and 3: one crucial piece of information is missing: does the expression of the mutants prevent the formation of clathrin-coated pits? It is absolutely imperative to quantify the number of pits (per surface area) in the WT or mutant expressing cells. It is also absolutely required to test if CME is indeed impaired in these cells by measuring the endocytosis of transferrin or the transferrin receptor.
- 4) In the same line, it is crucial to know if FCHO2 is still present at the clathrin pits, as it could compensate for the absence/deficiency of FCHO1
- 5) What is the "co-localization coefficient" used to quantify correlation with clathrin, and why did the authors apply different quantification methods to almost identical data sets?

FCHO1 regulates endocytosis and clustering of the T cell receptor (TCR)

- 1) l. 204: nowhere in ref 16 can be found data showing that the TCR:CD3 complex depends on clathrin to be internalised. Another more suitable reference has to be used.
- 2) It is of good practice to perform all, or at least the crucial experiments on more than just one CRISPR/Cas9 clone. Besides, a proof of the KO should be shown (western or flow).
- 3) Hopefully the images in figure 4a are not representative of those used for quantification: the sections (focal planes) chosen for the mutant or the KO are always somewhere in the cell where there is no cytoplasm (only nucleus). There is no intracellular compartment in the nucleus.

- 4) There is a very problematic confusion between clusters – which in T cells always designate aggregates of proteins within the plasma membrane – and intracellular compartment, which the authors here describe as aggregates or clusters. Internalised receptors end up in endosomes; they do not cluster in the cytoplasm. These are not clusters, they are TCR-positive endosomes. This has to be changed in l. 212-214 and 217. It also has to be changed in l. 244. Again, TCR clustering is something totally different (see work from Samelson, Dustin, and many others). Such confusion is a bit puzzling.
- 5) The quantification method use to count the “clusters” (aka TCR-positive endosomes) has to be described and documented. Especially because the image have a pretty bad signal to noise ration and overall because there seem to be a bias in how the focal planes are selected (see point 3) above)
- 6) As for SK-MEL cells, the author imperatively have to measure Tf or TfR endocytosis in the Jurkats (KO and +mutants). To show that CME is indeed impaired in T cells that have no FCHO1 or that express the mutant versions. Besides, the comparison with SK-MEL cells will show if FCHO1 mutation/deletion affects CME the same way in T cells and in another cell type. It would further allow to show if it is CME endocytosis in general that is dysfunctional in absence of FCHO1 or only TCR endocytosis.
- 7) TIRF – and not confocal - microscopy data showing the co-localisation of TCR and FCHO1 WT at the plasma membrane have to be shown.

Discussion

- 1) In l. 271, maybe the author should read the work of Wu et al., who show that the germinal centres have fewer B cells in clathrin light chain null mice (doi: 10.1073/pnas.1611189113). So the B cell defect might indeed be intrinsic.
- 2) Because it’s an essential part of the hypothesis illustrated in this manuscript, the authors have to check carefully the reference provided show internalisation of TCR through CME.
 - Ref. 26: Telerman et al. show that “after binding of OKT3, the complex (OKT3-T3) disappears rapidly from the cell surface. Using electron microscopy, we found that this down-regulation is due to the internalization of the complex”. No data showing that TCR is endocytosed through CME can be found in this study.
 - In ref 27, Dietrich et al only suggest that TCR is internalised through CME because immersing T cells in hypertonic medium prevents its internalisation. This paper is outdated, and we know now that quite a lot of things are as much, if not more disturbed than CME when cells are in an hypertonic medium. No direct and plausible data showing that TCR is endocytosed through CME can be found in this study.
 - Ref. 28: Boyer et al. only show that “staurosporine inhibits the receptor internalization induced by anti-TcR mAb by means other than inhibition of PKC”. No data showing that TCR is endocytosed through CME can be found in this study.
 - Ref. 29. Ohno et al. only show that TCR-CD3 interacts with clathrin-associated proteins. No data showing that TCR is endocytosed through CME can be found in this study.
 - And as mentioned above, no data showing that TCR is endocytosed through CME can be found in Ref. 16 either.
- 3) On the other hand, they fail to cite a reference showing that cholesterol depletion with methyl- β -cyclodextrin completely blocks TCR internalisation, which rather suggests a clathrin-independent process (10.1074/jbc.M409342200)

Reviewer #2:

Remarks to the Author:

In this manuscript the authors have identified six distinct homozygous mutations in seven unrelated pedigrees with variable T and B cell lymphopenia segregating with the disease in the gene encoding FCHO1, a protein implicated in the process of clathrin-mediated endocytosis (CME). Based on the established role of CME in the process of T cell activation and on the predicted deleterious effects of these mutations on FCHO1 function, the authors carried out a functional

characterization of these mutations by expressing the FCHO1 variants in a heterologous system (HEK293-T cells and Jurkat cells where the endogenous gene was deleted by CRISP/Cas9 editing), showing that the mutations result in mislocalization of FCHO1 at the plasma membrane and loss of interaction with its binding partners EPS1 and EPS1R. They also provide evidence that T cells from the patients are hyporesponsive to TCR engagement and fail to effectively internalize the TCR upon ligand binding. These defects were rescued by forced FCHO1 expression, underscoring the relevance of the mutations to the T cell defects in these patients.

The data are novel and contribute to our understanding of the pathways of vesicular traffic that have emerged as key components of the process of T cell activation. They additionally identify a new disease gene in the as yet large proportion of primary immunodeficiencies of unknown aetiology. There are however several issues that need to be addressed to consolidate the results, as detailed below.

Major points

Point 1. Figure 1. The authors describe the predicted outcome of the mutations on the transcripts based on bioinformatic analysis and subsequent measurement of the size of the transcripts. This result should be verified by sequencing the PCR products.

Point 2. Figure 2. The association of the wild-type FCHO1 puncta with the plasma membrane (lines 169-170) is not clear. The same applies to the pR679P and p.Stop687 μ HD mutants (lines 174-175). Since the mislocalization to the plasma membrane is a central feature of some of the mutants, the authors should strengthen these results by co-staining the cells with a plasma membrane marker (e.g. WGA). Additionally, immunoblot analysis of fractionated membranes would nicely complement the imaging data.

Point 3. Figure 2. Based on the co-localization analysis carried out by confocal imaging the authors conclude that "all mutants failed to interact with endogenous clathrin" (lines 177-178). This statement also applies to the corresponding figure legend. From these results the authors can only conclude that wild-type, but not mutant FCHO1, CO-LOCALIZE with clathrin. The INTERACTION with clathrin should be tested with other types of experiments, as a minimum by co-immunoprecipitation (which would complement the EPS1/EPS1R co-IP data presented in the same figure).

Point 4. The authors conclude that "FCHO1 deficiency impairs T cell development and responsiveness to TCR" (lines 262-263). To support this statement it is essential that the authors test these cells also for spontaneous as well as activation-induced cell death.

Point 5. Statistics for each type of analysis should be provided (significance is only shown for figures 4 and 6)

Point 6. Figure 5. The cytokine production data are not clear. The authors state that "FCHO1-deficient T cells produced considerably lower levels of IL-2 and IFN- γ . In contrast, secretion of TNF- α was comparable to healthy control cells, and IL-4 secretion was only marginally dependent of FCHO1 function (Fig. 5c)". First, it would seem that IL-4 secretion is comparable to healthy control cells, while TNF- α is marginally dependent on FCHO1. This is if one does the comparison against the healthy control. However, the results are different if the comparison is against mother and father, as they also produce significantly less IL-2, IFN- γ and TNF- α compared to healthy donors. Does this mean that mutation of one allele is sufficient to impair T cell activation? The authors should comment on this.

Point 7. Figure S2. The authors state that "point mutation in kindred E affects a splice donor and thereby results in a shortening of the FCHO1 by 51 amino acid encoded by exon 8 (lines 143-144).

However in panel (e) the differences between healthy donor and patient are not evident. The authors should add a sample from a heterozygous relative to clearly identify the two bands. Also, it is not clear why the E1 sample is run in triplicate. Is it the same sample or different samples from the same patient?

Pont 8. Discussion. Lines 331-334. The authors list the three possible outcomes of a defect in TCR internalization, including the opposite outcomes of signal amplification or attenuation. They should discuss their results in the context of these possible outcomes.

Minor points

Point 1. Figure 1 is mislabelled. There is no panel (b), at variance with the figure legend. The lettering of the panels should be changed accordingly in the description of the figure in the Results.

Point 2. In all immunoblots please use a line in addition to the molecular mass to precisely indicate the migration of each molecular mass marker

Point 3. Line 401, legend to figure 2; "In D arrows..." (D) should be in lowercase (d)

Point 4. Figure 4b, line 456. "Unstimulated controls are depicted in black". They are actually depicted in grey (with a black outline)

Point 5. The figures include a figure 6 and a supplemental figure 7 that are not part of the manuscript

Point 6. Figure S3. Do fibroblasts lack FCHO1 expression? This should be explicitly stated and the tissue specificity mentioned in the Introduction. Also, as mentioned above, the amplicons should be sequenced to confirm the predictions. In (a) several panels are truncated and they are all very fuzzy. In (e) does HD1/HD2 correspond to ctr1/ctr2 of panel (d)? If so they should have the same label.

Point 7. Figure S6. It is not clear why at longer time points the amount of FCHO1 and EPS1 co-immunoprecipitated with the N-flagged FCHO1 increases while the amount co-immunoprecipitating with the C-flagged FCHO1 decreases. Also for coherence with the other blots all molecular mass markers should be indicated in black.

Point 9. Line 430: how are "CD3 clusters" defined for the purpose of image analysis? Do they have a min-max size?

Point 10. Line 183. The authors refer to Sup. Movie 1 (also in the respective legend), but these are actually 3 movies.

Reviewer #3:

Remarks to the Author:

The manuscript from Klein and co-workers describes their identification of six (and functional analysis of three) FCHO1 mutations in humans.

The authors convincingly show an essential role for FCHO1 in clathrin-mediated endocytosis in T-cell function. This is an important result especially if the improvements mentioned below can be incorporated into the paper.

Mutant FCHO1 proteins are affected in the F-BAR or μ HD regions and are thus expected to be compromised in either plasma membrane localisation or in their interaction with binding partners of FCHO1 μ HD respectively. The μ HD mutants R679P and Stop687 are shown to be severely perturbed in their ability to immuno-precipitate the FCHO1 binding partners EPS15 and EPS15R (Fig 2a). These conclusion are nicely supported.

* The appearance of GFP-FCHO1-A34P in Fig2d and Sup Fig5 (large aggregates dissociated from the plasma membrane) is not consistent with the appearance of the same protein in Fig 2b & Fig 2c (well-defined punctae that appear to co-localise with EPS15 and adaptin, respectively). The reason for this discrepancy is not made clear by the authors.

In agreement with Fig 2a the μ HD mutants R679P and Stop687 do not co-localise with EPS15 and adaptin (Fig 2b, 2c).

* Worryingly, the images of RFP-clathrin in Fig 2d do not appear to be of acceptable quality to justify interpretations or quantitation (showed in Fig 2e).

* The results presented in Fig 3a are also not of a quality to permit interpretation. Neither the movies nor the panels presented as Fig 3a are convincing. The quality of imaging needs to be improved.

Thus far, the data show the μ HD mutants R679P and Stop687 are impaired in interaction with binding partners and are thus mis-localised. However, interpretation of the functional outcome of the A34P mutation is not clear.

Given that humans harbouring FCHO1 mutations display T cell defects, the authors then examine the possibility that FCHO1 may be involved in internalisation of the CD3:TCR complex. The data presented in Fig 4 are convincing and suggest that FCHO1 indeed plays a role in internalisation of the complex. The authors then go on to examine possible interaction between FCHO1 and CD3 molecules that possess putative μ HD interaction motifs –

* however, the experiments presented in Sup Fig 6 have not been sufficiently described in the Legend for this reviewer to be able to participate in its interpretation.

Finally, the authors demonstrate that FCHO1 modulates T cell proliferation and cytokine secretion in primary human cells (Fig 5).

* Note: the submitted manuscript also contains an additional figure (Fig 6) but no mention has been made of this figure in the Legends or in the text.

In summary, the manuscript should be significantly improved before publication. The improvements suggested above, if incorporated, would make the manuscript suitable for publication because the authors present a very interesting aspect of FCHO1 biology. Also, it would be informative for the authors to comment on why FCHO2 is not able to compensate for the defects reported in this manuscript.

Reviewer #4:

Remarks to the Author:

The manuscript is in general well written. The authors link homozygous deleterious mutations in FCHO1 to T and B cell lymphopenia. In doing so they make a strong case for the involvement of FCHO1 in development and function of T cells. One interesting aspect of the discovery is the diversity of the disruptive mechanism observed across the seven families via missense, nonsense, splice altering and frameshift changes. The authors have functionally validated the impact of half of these mutations. In particular the splice altering mutations are only shown to impact the protein

length. Inclusion of the binding properties or sub-cellular localisation assays for these mutants would further improve the study.

The inclusion of a negative result in Suppl. Fig. 6 is highly commendable. In general the discussion section provides a fair and balanced overview of the current state of knowledge regarding FCHO1 and the additional gains and future directions defined by the current study. The authors note the severe malformation observed as a result of knockdown experiments in zebrafish and other model organisms in previous studies. They mention potential evolutionary change in functional role of FCHO1 in humans. Although this is plausible, it is not clear if the mutants observed in the current study lead to a complete loss of function. That would provide an alternate hypothesis as well.

Minor points

Figure 1 panels likely underwent some last minute changes and hence the labels do not align with the references in the manuscript

Figure 6 legend is missing

Whole exome sequencing and variant filtering: details for families E and G seem to be missing. Also it would be helpful to include the polyphen2 prediction for the missense mutations in this section.

Point to Point replies to the reviewers

**Reviewer #1 (Remarks to the Author):**

*This is an interesting manuscript showing the requirement of the clathrin-mediated*
*endocytosis effector FCH domain only 1 for T cell receptor (TCR) internalisation. The strong*
*aspect of the study is the link between FCHO1 mutations and lymphopenia in human. The*
*main weakness of the work is the fact that these mutations, which totally prevent FCHO1 to*
*associate/initiate with clathrin-coated pits, have an effect only on TCR endocytosis (as*
*mentioned in the discussion lines 292-295 and 306-308). It is a source of concern because*
*the data presented do not allow figuring out if the expression of the mutant prevents clathrin-*
*mediated endocytosis (CME) at all, or just the CME of the few proteins (including TCR) that*
*require FCHO1 to get into clathrin coated pits. The authors even touch that issue, l.219-220,*
*suggesting that the absence of FCHO1 may impact the whole organisation of T cell plasma*
*membrane. Hence, the defect in T cells activation could be due to a disorganisation of the*
*plasma membrane, and not to a defect in TCR internalisation.*

*1. For these reasons, the authors ABSOLUTELY have to provide an example of a receptor,*
*known to be internalised through CME, which is not affected by the absence of FCHO1. It*
*would also have more impact if these cells come from the patients.*

We have addressed the reviewer's concerns. Of note, in contrast to the reviewer's
interpretation, we have never stated that "...mutations, which totally prevent FCHO1 to
associate/initiate with clathrin-coated pits, ...". Since it is of critical relevance whether FCHO1
controls CME globally or not, we wish to reiterate what we have written in the original
manuscript:

Discussion, lines 367-370:

"Given the central role of FCHO1 for the formation of clathrin coated pits and endocytosis it is
surprising that FCHO1 deficiency results in T-cell immunodeficiency rather than more global
defects of development. In contrast, Caenorhabditis elegans deficient for FCHO show body
malformation and uncoordinated locomotion (Hollopeter, Lange et al. 2014)."

Discussion, lines 380-382:

"...the observation that a) two FCHO paralogs have emerged (Dergai, Iershov et al. 2016)
and b) FCHO may act as receptor-specific adaptors (e.g. BMP-mediated signaling in zebra
fish) rather than universally important initiators of clathrin-mediated endocytosis (Umasankar,
Sanker et al. 2012)."

Regardless of this misunderstanding, we fully agree with the reviewer that the relationship
between FCHO1 mutants and global effect on clathrin-mediated endocytosis (CME) of
various receptors was not fully addressed in our original manuscript. Hence, following the
reviewer's advice, we have done extensive additional experiments to address this question:

a) VSV-G pseudotyped virions enter cells via clathrin-dependent endocytosis (Daecke,
Fackler et al. 2005, Popov, Strack et al. 2011). We took advantage of VSV-G
pseudotypes of HIV-1 virus and performed a series of experiments to address the
question whether VSV-G pseudotyped HIV-1 virions are able to infect FCHO1-
knockout Jurkat cells (six clones tested), in comparison to WT cells (five clones
tested). We did not detect any differences between WT and FCHO1-knockout Jurkat
clones regarding HIV-1 fusion and entry, indicating that this process is independent of
FCHO1 and initiation of CME via this membrane nucleating adaptor (revised Fig. 4e, f
and Sup. Fig. 10-11).

b) To further substantiate our notion that FCHO1-mutations affect internalisation of only
some (and not all) membrane-bound proteins, a set of experiments was performed on
patient-derived fibroblasts. The transferrin receptor (TfR) is a classical example of a
CME-dependent receptor. Upon ligand binding, transferrin (Tf) is internalised through
CME (Pearse and Robinson 1990, Miller, Shipman et al. 1991, Liu, Aguet et al. 2010,
Mayle, Le et al. 2012). Hence, we followed internalisation of TfR upon binding to
fluorescently labelled Tf in fibroblasts from 2 patients from 2 independent families and
healthy donor fibroblasts. Internalisation of TfR was unaffected in cells which express
mutated FCHO1 (revised Figure 4g), supporting our hypothesis that this membrane
nucleating protein may have evolved to specifically control internalisation of
lymphocyte-specific receptors, like TCR.

Together, both sets of independent experiments document that FCHO1-deficiency does not
globally affect CME.

*III/ In the same line, the authors have to perform an experiment to show that the general*
*organisation of the plasma membrane is not affected in T cells (Jurkats) that lack FCHO1. For*
*instance using FCS or image correlation. Or at least, to show that there no difference in the*
*formation of TCR microclusters on antibody-coated glass (the Samelson way) or on*
*supported bilayers (the Dustin way).*

We agree with the reviewer, that measurements of TCR microclusters on coated glass or on
supported bilayers would be very interesting. However, this technology is highly specialized
and only few laboratories in the world have the necessary equipment and expertise. Our
paper does not address a comprehensive mechanistic picture of the processes at the plasma
membrane – but it provides novel and conclusive data on the dysfunction of FCHO1 in
patients with FCHO1 mutations and lymphocytes deficiency.

Regardless of these limitations, we are able to provide additional morphological data on the
organisation of the plasma membrane in FCHO1-deficient cells. We have stained SK-MEL-2
cells (devoid of endogenous FCHO1) which were transiently transfected with WT and 3
different mutants of FCHO1 (example for F-BAR domain mutant and μ HD domain mutants)
with fluorescently labelled wheat germ agglutinin (WGA). WGA is one of the most widely
used lectins in cell biology, which binds to glycoconjugates present in the plasma membrane,
hence is a good tool to study the morphology of the plasma membrane. These experiments
did not reveal any gross aberrations in the architecture of the plasma membrane (new data
are now shown in Sup. Fig. 5).

While it is beyond the scope of this manuscript to determine TCR microclusters and the
resolution of FCHO1 function in submicroscopic dimensions, our findings provide solid
evidence to link the development and function of T cells to the function of FCHO1.

*In addition, many relevant control experiments are missing (see specific comments).*
*More specific comments:*
*Effects of patient-associated mutations on FCHO1 function*
*1) There is some confusion in figure 2. The cells are supposed to express Clathrin-RFP (l.*
*166). Yet, the red channel shows EPS15 in b, adaptin in c and indeed clathrin in d. Legends*
*and text have to be amended.*

We regret that our data presented in original Figure 2 caused some confusion. All colours
displayed on the images are pseudo-colours and hence, colour choice is arbitrary. In the

original version of the manuscript, the information about colour coding was provided on the
right side of each panel: b) EPS15 (detected by specific antibody, visualised by fluorescently
labelled secondary antibody as described in MM section, l. 857-858 of revised manuscript);
c) adaptin (detected by specific antibody, visualised by fluorescently labelled secondary
antibody as described in MM section, l.859-860 of revised manuscript); d) endogenous
clathrin light chain A, of which one allele has been coupled to RFP as described in MM
section , l.743-745 of revised manuscript).

In our revised version of the paper, we add the information about colour choices to the
respective figure legends (line 494-495 of revised manuscript).

*2) l. 171-172, the images do not really allow to see if the punctate structures in the F-BAR*
*domain mutant are dissociated from the plasma membrane. A xz or yz image of those cells is*
*required to support this statement.*

To provide spatial views, we have repeated the experiment and now provide z-stacks that
unequivocally document that all FCHO1-mutants are dissociated from the plasma membrane
(see Sup. Fig. 5 and 6). The wording in the text of our revised manuscript has been adjusted
accordingly, lines: 172-175; 198-200; 636-639.

*3) Figure 2 and 3: one crucial piece of information is missing: does the expression of the*
*mutants prevent the formation of clathrin-coated pits? It is absolutely imperative to quantify*
*the number of pits (per surface area) in the WT or mutant expressing cells. It is also*
*absolutely required to test if CME is indeed impaired in these cells by measuring the*
*endocytosis of transferrin or the transferrin receptor.*

The elucidation of the detailed spatiotemporal organisation of FCHO1 on the formation of
CME goes far beyond the scope of this manuscript. In our revised manuscript, we provide
additional data that unequivocally prove a critical dysfunction of FCHO1 mutants which is
linked to aberrant subcellular distribution of the mutants and defective TCR internalisation.
We clarify in our revised version, that FCHO1-deficiency does not result in a global and
complete dysfunction of CME, since VSV-G-induced virus uptake and internalisation of
transferrin receptor is not perturbed in FCHO1-deficiency (see revised Figures 4e-g and Sup.
Fig. 10 and 11). The text been adjusted in the revised version of the manuscript, lines: 196-
136 200; 276-277; 284-286; 383-387;

*4) In the same line, it is crucial to know if FCHO2 is still present at the clathrin pits, as it could*
*compensate for the absence/deficiency of FCHO1*

We agree with reviewer 1 that it would be interesting to know whether FCHO2 could, at least
in part, compensate for the absence of FCHO1. Please note however, that both FCHO
paralogues act as homodimers, and differ in their abilities to bind adaptor proteins. Although
both FCHO1 and FCHO2 can trigger CME, it is unlikely that FCHO2 fully compensates for a
lack of function of FCHO1.

Moreover, to gain knowledge whether “FCHO2 is still present at the clathrin pit”, one would
have to specifically stain FCHO2 – which is currently not possible in light of a lack of suitable
monoclonal antibodies. In the past, renowned experts (such as Dr. Harvey McMahon from
MRC LMB laboratory in Cambridge) have raised concerns with respect to interpreting the
function of FCHO1/2 whenever the expression levels are not corresponding to physiological
levels (Henne, Boucrot et al. 2010). We took this concern seriously. Hence, in imaging
experiments we paid meticulous attention to restore physiological expression levels of
FCHO1 (e.g. by knock-in into the physiological genomic locus. Transient transfection of DNA

encoding a GFP-FCHO1 fusion protein results in overexpression and non-physiological
conditions. While this may be useful to address some scientific questions (such as coarse
subcellular localisation), it clearly does not recapitulate the physiological function. In view of
these concerns, we are hesitant to rely on ectopic expression of a tagged-FCHO2 protein for
these fine-tuned analyses.

*5) What is the “co-localization coefficient” used to quantify correlation with clathrin, and why*
*did the authors apply different quantification methods to almost identical data sets?*

We apologize for not having explained our quantification methods in sufficient detail. The
statistical method to determine the co-localization coefficient utilised in Fig 3. depends on the
biochemistry and stoichiometry.

a) Whenever the interaction of FCHO1 with its downstream partners occurs in a
stoichiometric ratio, i.e. one μ HD domain of FCHO1 interact with one EPS15
molecule, one FCHO1 molecule and AP-2 (adaptin subunit), we quantified these
interactions by means of the **Pearson correlation** analysis, which describes the
correlation of the intensity distribution between channels.
- b) Whenever the association between FCHO1 and partners is indirect or without any
stoichiometric relationship, we made use of **co-localisation coefficients** parameters.
For example, we asked how many green pixels (FCHO1) colocalise with red pixels
(clathrin) regardless of the pixel's intensity. For details please refer to (Zinchuk and
Zinchuk 2008).

*FCHO1 regulates endocytosis and clustering of the T cell receptor (TCR)*
*1) l. 204: nowhere in ref 16 can be found data showing that the TCR:CD3 complex depends*
*on clathrin to be internalised. Another more suitable reference has to be used.*

We are grateful to the reviewer for indicating this mistake. Indeed, reference #16 is not the
right source of information to support the statement in the result section lines 208-209
(revised manuscript). References #19, #20, #21, #22 (numbering corresponds to revised
manuscript) as well as new reference #23 (Dietrich, Kastrup et al. 1997) and #24 (review by
(Balagopalan, Barr et al. 2009) have been introduced instead of reference #16 in the revised
version of the manuscript.)

*2) It is of good practice to perform all, or at least the crucial experiments on more than just*
*one CRISPR/Cas9 clone. Besides, a proof of the KO should be shown (western or flow).*

We completely agree with the reviewer. In general, all experiments presented in the original
version of the manuscript have been done using multiple clones, in most cases a minimum
three independent clones (both wt and ko), as indicated in figure legend (lines 540-541 of
revised manuscript) and MM section (lines 786-788.)

We concur with the reviewer that it is of good practice to corroborate the results of genetic
manipulation. In the revised version of the manuscript we included additional Sanger
sequencing data which show bi-allelic out-of-frame mutation in multiple clones (Sup. Fig.
12).

*3) Hopefully the images in figure 4a are not representative of those used for quantification:*
*the sections (focal planes) chosen for the mutant or the KO are always somewhere in the cell*
*where there is no cytoplasm (only nucleus). There is no intracellular compartment in the*
*nucleus.*

In Figure 4a, the focal sections were acquired the same way for WT and KO Jurkat clones. It is virtually impossible to look at the cytoplasm in cells in which the nucleus absorbs almost the entirety of the intracellular space (like in the case of any lymphocyte). To improve visualization, we prepared a new Supplementary Figure (Sup. Fig. 8), where we present Z-stack used for 3-D reconstruction and 6 selected optical sections covering the volume of the Jurkat cells from bottom to the top.

4) There is a very problematic confusion between clusters – which in T cells always designate aggregates of proteins within the plasma membrane – and intracellular compartment, which the authors here describe as aggregates or clusters. Internalised receptors end up in endosomes; they do not cluster in the cytoplasm. These are not clusters, they are TCR-positive endosomes. This has to be changed in l. 212-214 and 217. It also has to be changed in l. 244. Again, TCR clustering is something totally different (see work from Samelson, Dustin, and many others). Such confusion is a bit puzzling.

In our view, the reviewer brings up a semantic issue. The nomenclature of “clusters” and “aggregates” are often used interchangeably. We do not refer to supramolecular signalling complexes on nanoscale, but to multivalent TCR:CD3 molecules which are formed upon TCR stimulation and can be detected by light microscopy.

To support our point of view, we refer to a review from the laboratory of L. Samelson (Sherman, Barr et al. 2013), p.21): “Multi-molecular signaling complexes drive the earliest events of immune cell activation via immunoreceptors with unexplained specificity and speed. Fluorescence microscopy has shown that these complexes form microclusters at the plasma membrane of activated T cells upon engagement of their antigen receptors (TCRs). Although crucial for cell function, much remains to be learned about the molecular content, fine structure, formation mechanisms, and function of these microclusters. Recent advancements in super-resolution microscopy have enabled the study of signaling microclusters at the single molecule level with resolution down to ~20nm.”

Another work from (Alarcon, Swamy et al. 2006), p.490 supports the fact that TCR clusters or microclusters can be visualised by light microscopy after stimulation and receptor aggregation and this refers to multivalent TCR:CD3 molecules.

Such complexes (clusters) are subsequently internalised to intracellular compartments, like endosomes. The authors remain unconvinced about why after TCR:CD3 cluster internalisation from the plasma membrane the name of the primary structure – called “cluster” - should be changed to “aggregate” or “TCR-positive endosomes” as indicated in reviewer’s comment. We would like to leave this decision to the editor.

5) The quantification method use to count the “clusters” (aka TCR-positive endosomes) has to be described and documented. Especially because the image have a pretty bad signal to noise ration and overall because there seem to be a bias in how the focal planes are selected (see point 3) above)

The quantification method related to data presented on Figure 4 has been described in the legend to Figure 4 and in MM section of the original version of our manuscript, please see lines 526-530 and 970-971. In brief, visible CD3 clusters/aggregates per cells were counted and averaged for each random field of view. Hence, each point on Fig 4a (quantification) indicates an average number of clusters per cell that could be found in one field of view. To clarify any misconception, we have included, in our revised paper, optical sections from Z-stacks (Sup. Fig. 8, see also our answer to comment #3).

6) As for SK-MEL cells, the author imperatively have to measure Tf or TfR endocytosis in the
Jurkats (KO and +mutants). To show that CME is indeed impaired in T cells that have no
FCHO1 or that express the mutant versions. Besides, the comparison with SK-MEL cells will
show if FCHO1 mutation/deletion affects CME the same way in T cells and in another cell
type. It would further allow to show if it is CME endocytosis in general that is dysfunctional in
absence of FCHO1 or only TCR endocytosis.

Following the reviewer's advice, we have performed new experiments to quantify
internalization of transferrin receptors in WT and FCHO1 mutant cells from the patient (see
new Fig. 4g). As pointed out above, we conclude that even in the absence of FCHO1, CME
endocytosis is not globally affected. To provide further proof of this notion, we additionally
performed experiments on Jurkat clones, where we tracked the entry of VSV-G HIV-1, which
is clathrin dependent. Also, in this case, virus entry in FCHO1-KO Jurkat clones was not
perturbed as compared to WT clones, see also answer to general comment.
Hence, it is plausible that deficiency in FCHO1 specifically affects TCR endocytosis and
which result in perturbed T-cell development/activation, as seen in our patients.

7) TIRF – and not confocal - microscopy data showing the co-localisation of TCR and
FCHO1 WT at the plasma membrane have to be shown.

Whereas, it would be certainly interesting to employ TIRF technique, is far beyond the scope
of this manuscript. Moreover, confocal microscopy has been the main technique used in the
original paper identifying FCHO1 as membrane nucleating protein initiating clathrin mediated
endocytosis (Henne, Boucrot et al. 2010).

Discussion

1) In l. 271, maybe the author should read the work of Wu et al., who show that the germinal
centres have fewer B cells in clathrin light chain null mice (doi: 10.1073/pnas.1611189113).
So the B cell defect might indeed be intrinsic.

We are grateful for pointing out this important work, however, it does not justify any
conclusions on B-cell intrinsic consequences of clathrin-deficiency.

Wu et al. use clathrin light chain conditional knockout mice, where exon 1 of clathrin light
chain A locus is flanked by LoxP sites and such animals are crossed with mice expressing
Cre recombinase under transcriptional control of ACTB (actin) gene promoter. This results in
global (not B cell specific) deletion of clathrin light chain, and half of the homozygous mice
died within a week after birth.

Furthermore, germinal centre reaction is a complex process, involving follicular dendritic cells
which present antigen to T helper cells, which in turn provide help to B-cells during affinity
maturation. All mentioned cell types in mice used in this publication are affected by the lack
of clathrin light chain – not only B-cells.

2) Because it's an essential part of the hypothesis illustrated in this manuscript, the authors
have to check carefully the reference provided show internalisation of TCR through CME.
- Ref. 26: Telerman et al. show that "after binding of OKT3, the complex (OKT3-T3)
disappears rapidly from the cell surface. Using electron microscopy, we found that this down-
regulation is due to the internalization of the complex". No data showing that TCR is
endocytosed through CME can be found in this study.

The paper by Telerman et al. uses electron microscopy data to study internalization of TCR
receptor upon OKT3 (anti CD3) stimulation. The data where generated in the year 1987,

when access to quality controlled specific monoclonal antibodies, cloning techniques and
sophisticated high-end fluorescent microscopy was still limited. Nevertheless, the authors
with confidence identified TCR being internalized through clathrin coated pits, based on data
presented on Figure 2a of the cited paper.

- In ref 27, Dietrich et al only suggest that TCR is internalised through CME because
immersing T cells in hypertonic medium prevents its internalisation. This paper is outdated,
and we know now that quite a lot of things are as much, if not more disturbed than CME
when cells are in an hypertonic medium. No direct and plausible data showing that TCR is
endocytosed through CME can be found in this study.

We agree with the reviewer that reference #27 (#20 in revised manuscript) (dated 1994) is
not the newest one and experiments with hypertonic medium, even though state of the art at
the time, are not sufficient to provide a definitive conclusion. Hence, the authors carefully
state: p.2157: “PKC induced TCR down-regulation is probably mediated by endocytosis via
clathrin coated pits”. Furthermore, Dietrich et al. is not the only paper that we cite in support
of our statement that TCR endocytosis depends on CME (see references #19, 21, 22, 23, 24
of the revised paper). Unfortunately, there is no specific inhibitor of CME available.
Chlorpromazine is certainly not specific, Pitstop2 is also not specific. Nevertheless,
chlorpromazine has been used by the scientific community as an agent that blocks CME
(Willox, Sahraoui et al. 2014). We are aware of all these intrinsic limitations. In our original a
paper we wrote (1.401-402): “The TCR can be internalized via clathrin-dependent (Telerman,
Amson et al. 1987, Boyer, Auphan et al. 1991, Dietrich, Hou et al. 1994, Ohno, Stewart et al.
1995) or clathrin-independent mechanisms (Compeer, Kraus et al. 2018)”

This is a cautious statement which is not in contrast to the reviewer’s opinion.

- Ref. 28: Boyer et al. only show that “staurosporine inhibits the receptor internalization
induced by anti-TcR mAb by means other than inhibition of PKC”. No data showing that TCR
is endocytosed through CME can be found in this study.

We are not sure why reviewer #1 is concerned with the reference Boyer et al. (#21 revised
version). Boyer et al provide data (based on electron microscopy studies, similar to
(Telerman, Amson et al. 1987) which demonstrate that TCR is internalized through “coated
pits” (see below):

Results, chapter 3.1, p.1626: “Evidence from electron microscopy for TcWCD3 internalization
via **coated pits** in CTL clone KB5.CU: comparison with internalization of H-2”

Figure 1: “Electron micrographs showing entry of TcR into b KB5.C20 cells. Entry of the TcR
was monitored using mAb DBsirt-1 followed by protein A-gold. (a) Surface labeling at 0°C.
After 5 (b) and 10 (c) min at 37°C, some labeling was observed **in coated pits and coated
vesicles.**”

Discussion, p.1632: “Available data suggest a sequence of three distinct steps controlling
TcR/CD3 internalization/sequestration induced by anti-TcR mAb (see Fig. 10). The first step,
common to receptor cycling and sequestration, involves the association with **adaptor
proteins in coated pits** [7] which requires the presence of the appropriate aromatic residue
[7] in the cytoplasmic domain of one of the CD3 components, **as shown for the transferrin
receptor** [49].”

The authors refer to “coated pits” in reference #21 and cite work of Pearse BMF, EMBO
1988, p.3331, where “coated” means:” *The coat structure is composed of two distinct*
*structural units, the clathrin triskelion and a complex containing 100 and 50 kd*
*polypeptides.”*

Hence, Boyer et al. provide experimental data that TCR is internalized through “coated pits”,
through which authors understand “clathrin coated pits”.

In summary, we do not see any reason to change our statement (original manuscript, l.322-
323: “*The TCR can be internalized via clathrin-dependent (Telerman, Amson et al. 1987,*
*Boyer, Auphan et al. 1991, Dietrich, Hou et al. 1994, Ohno, Stewart et al. 1995) or clathrin-*
*independent mechanisms(Compeer, Kraus et al. 2018)”*

- Ref. 29. Ohno et al. only show that TCR-CD3 interacts with clathrin-associated proteins. No
data showing that TCR is endocytosed through CME can be found in this study.
- And as mentioned above, no data showing that TCR is endocytosed through CME can be
found in Ref. 16 either.

The reviewer is right that in reference #43 (revised version) by Ohno et al., there are not
experiments showing that TCR is endocytosed through CME, but that TCR:CD3 interacts
with clathrin associated proteins. However, on the page 1873, Ohno et al. state: “*In contrast*
*to tyrosine-based signals, the di-leucine-based signal DKQTLL from CD3-gamma, which also*
*functions as an internalization and lysosomal-targeting signal...”*

Together, this information seems to be appropriate to support the statement made in the
discussion of our submitted manuscript, l.322-323: “*The TCR can be internalized via clathrin-*
*dependent (Telerman, Amson et al. 1987, Boyer, Auphan et al. 1991, Dietrich, Hou et al.*
*1994, Ohno, Stewart et al. 1995) or clathrin-independent mechanisms(Compeer, Kraus et al.*
*2018)”*.

The explanations regarding the choice of reference #16 has been made in previous comment
to the reviewer: “FCHO1 regulates endocytosis and clustering of the T cell receptor (TCR),
number 1)”

3) On the other hand, they fail to cite a reference showing that cholesterol depletion with
methyl-B-cyclodextrin completely blocks TCR internalisation, which rather suggests a
clathrin-independent process (10.1074/jbc.M409342200)

We are grateful for mentioning this important work, which we include in the revised version of
the manuscript. However, as pointed out above, we have cautiously stated:

discussion of our submitted manuscript, l.401-402: “*The TCR can be internalized via clathrin-*
*dependent (Telerman, Amson et al. 1987, Boyer, Auphan et al. 1991, Dietrich, Hou et al.*
*1994, Ohno, Stewart et al. 1995) or clathrin-independent mechanisms(Compeer, Kraus et al.*
*2018)”*

**Referee #2: immunodeficiencies, T cell biology**

**Reviewer #2 (Remarks to the Author):**

*In this manuscript the authors have identified six distinct homozygous mutations in seven*
*unrelated pedigrees with variable T and B cell lymphopenia segregating with the disease in*
*the gene encoding FCHO1, a protein implicated in the process of clathrin-mediated*
*endocytosis (CME). Based on the established role of CME in the process of T cell activation*
*and on the predicted deleterious effects of these mutations on FCHO1 function, the authors*
*carried out a functional characterization of these mutations by expressing the FCHO1*
*variants in a heterologous system (HEK293-T cells and Jurkat cells where the endogenous*
*gene was deleted by CRISP/Cas9 editing), showing that the mutations result in*
*mislocalization of FCHO1 at the plasma membrane and loss of interaction with its binding*
*partners EPS1 and EPS1R. They also provide evidence that T cells from the patients are*
*hyporesponsive to TCR engagement and fail to effectively internalize the TCR upon ligand*
*binding. These defects were rescued by forced FCHO1 expression, underscoring the*
*relevance of the mutations to the T cell defects in these patients.*

*The data are novel and contribute to our understanding of the pathways of vesicular traffic*
*that have emerged as key components of the process of T cell activation. They additionally*
*identify a new disease gene in the as yet large proportion of primary immunodeficiencies of*
*unknown aetiology. There are however several issues that need to be addressed to*
*consolidate the results, as detailed below.*

We appreciate the overall assessment of reviewer #2.

*Major points*

*Point 1. Figure 1. The authors describe the predicted outcome of the mutations on the*
*transcripts based on bioinformatic analysis and subsequent measurement of the size of the*
*transcripts. This result should be verified by sequencing the PCR products.*

We agree with the reviewer that sequencing of transcripts is helpful to validate the predicted
outcome. This is particularly relevant in pedigree E, F (splice site mutation). To address this
issue, we have included Sanger sequencing results of cDNA, confirming our predictions. See
Sup. Fig 2c (lower panel) and Sup. Fig. 3e of the revised manuscript.

*Point 2. Figure 2. The association of the wild-type FCHO1 puncta with the plasma membrane*
*(lines 169-170) is not clear. The same applies to the pR679P and p.Stop687 μ HD mutants*
*(lines 174-175). Since the mislocalization to the plasma membrane is a central feature of*
*some of the mutants, the authors should strengthen these results by co-staining the cells*
*with a plasma membrane marker (e.g. WGA). Additionally, immunoblot analysis of*
*fractionated membranes would nicely complement the imaging data.*

We agree with the reviewer that this statement in our original manuscript was not entirely
substantiated by experimental data. It is indeed pivotal to determine the cellular localisation
of the F-BAR domain mutant. Co-localisation of the μ HD domain mutants is of lesser
concern, as dysfunction of the μ HD domain results in inability to interact with binding partners
(EPS15 and EPS15R as shown in WB).

Following the reviewer's suggestion, we have performed additional experiments and stained
the plasma membrane with fluorescently labelled wheat germ agglutinin (WGA) (see Sup.
Fig 5). Only WT FCHO1 correlates well with plasma membrane marker WGA, whereas all
mutants appear as dissociated from the plasma membrane. F-BAR mutants form large
clusters distributed throughout the cell. Two μ HD domain mutants fail to form any clusters,
and the fluorescence signal is distributed throughout entire cell (except the nucleus). We

point out that there is a caveat with respect to signal intensity: FCHO1 μ HD-mutants fail to
form productive clusters and therefore the dim signals cannot faithfully be detected (l. 189-
190).

We agree with reviewer 2 that immunoblot analyses of fractionated membranes would nicely
complement the imaging data. However, we feel that these studies are not critical to
substantiate our claims especially in view of our new data on WGA staining and co-
localisation of F-BAR mutant FCHO proteins.

*Point 3. Figure 2. Based on the co-localization analysis carried out by confocal imaging the*
*authors conclude that "all mutants failed to interact with endogenous clathrin" (lines 177-*
*178). This statement also applies to the corresponding figure legend. From these results the*
*authors can only conclude that wild-type, but not mutant FCHO1, CO-LOCALIZE with*
*clathrin. The INTERACTION with clathrin should be tested with other types of experiments,*
*as a minimum by co-immunoprecipitation (which would complement the EPS1/EPS1R co-IP*
*data presented in the same figure).*

We agree with the reviewer and therefore we changed our wording accordingly. In the
revised version of our manuscript, l. 177-178 , the sentence reads as follows:

*"Importantly, all mutants failed to colocalize with endogenous clathrin (Fig. 2d-e)."*

The reviewer is right that only co-immunoprecipitation experiments would conclusively
support the statement about direct protein-protein interaction. However, it is known that
FCHO1 does not directly interact with clathrin. FCHO1 acts at the very early stage of CCP
formation by facilitating membrane invagination via binding to EPS15 and EPS15R. Of note,
we could also document this interaction (see Figure 2a). Clathrin acts at the later stage of
cargo selection and is recruited to the complex via another adaptor (AP2), which also does
not directly interact with FCHO proteins (McMahon and Boucrot 2011). Hence, further co-
immunoprecipitation experiments would not be conclusive in this context.

*Point 4. The authors conclude that "FCHO1 deficiency impairs T cell development and*
*responsiveness to TCR" (lines 262-263). To support this statement it is essential that the*
*authors test these cells also for spontaneous as well as activation-induced cell death.*

We agree with reviewer 2 that it would be interesting to perform studies of spontaneous and
activation-induced cell death of cells expressing mutant FCHO1. However, there are a
number of limitations:

Development of T cells takes place in the thymus, hence susceptibility to apoptosis of
FCHO1 deficient developing thymocytes cannot easily be tested in peripheral T cells. For
peripheral T cells, lack of responsiveness to TCR does not necessary result in immediate
apoptosis. In fact, upon TCR stimulation, T cells of one FCHO1-deficient patient (C1, Fig.
5b) almost completely failed to proliferate. Nevertheless, T cells were able to produce
cytokines (Fig. 5c). This suggests incomplete T-cell responses rather than TCR-induced
apoptosis. Finally, access to primary FCHO1-deficient cells is extraordinarily challenging.
Some patients underwent allogeneic hematopoietic stem cell transplants, others live in
remote and quite inaccessible areas such as the Gaza Strip. Others are currently undergoing
various immunomodulatory treatments, including chemotherapy.

For all these reasons, experiments with primary peripheral blood T cells from patients are
very hard to do and not likely to yield meaningful results.

Nevertheless, we have made an effort to address the reviewer's concerns. We chose an in
vitro co-culture model of T cell development and assayed the importance of CME by using a
pharmacological inhibitor of CME, chlorpromazine (see new data in Fig 6). Chlorpromazine

inhibits CME by preventing the assembly and disassembly of clathrin lattices on cell surfaces
and on endosomes (Wang, Rothberg et al. 1993). We are aware that pharmacological
inhibitors are hardly ever specific. In this perspective, also chlorpromazine is not a truly
specific inhibitor, however there is no alternative reagent available (Pitstop 2 has been
shown to be unspecific and our experiments with Pitstop 2 and control reagents from the
same company were inconclusive). Therefore, we interpret our new data presented in new
Fig. 6 with caution. We took advantage of OP9-DL1 co-culture system where we followed
development of T-cells from (CD4 CD8 double negative) DN3 sorted murine precursors in
the presence and absence of chlorpromazine. Data presented in Fig. 6 indicate that
development of T-cells is delayed upon pharmacological inhibition of CME. Given that DN3-
to-DP (CD4 CD8 double positive) transition strictly depends on quality of signal delivered by
preTCR, and the fact that all other lineage tested (B-lymphocytes and myeloid cells) are
refractory to chlorpromazine in this concentration, this experiment supports our notion that
CME is relevant during T cell development.

*Point 5. Statistics for each type of analysis should be provided (significance is only shown for*
*figures 4 and 6)*

We thank the reviewer for this comment. We added statistical analysis whenever appropriate
in the revised version of the manuscript. Please see Fig. 2e, Fig. 3c and Fig. 6a-d.

*Point 6. Figure 5. The cytokine production data are not clear. The authors state that*
*"FCHO1-deficient T cells produced considerably lower levels of IL-2 and IFN- γ . In contrast,*
*secretion of TNF- α was comparable to healthy control cells, and IL-4 secretion was only*
*marginally dependent of FCHO1 function (Fig. 5c)". First, it would seem that IL-4 secretion is*
*comparable to healthy control cells, while TNF- α is marginally dependent on FCHO1. This is*
*if one does the comparison against the healthy control. However, the results are different if*
*the comparison is against mother and father, as they also produce significantly less IL-2,*
*IFN- γ and TNF- α compared to healthy donors. Does this mean that mutation of one allele is*
*sufficient to impair T cell activation? The authors should comment on this.*

We agree with the reviewer that the cytokine production data deserve clarification and better
explanation. The parents produce less IL-2, IFN- γ and TNF- α in comparison to healthy
donors. This finding is currently not easy to explain. It is consistent with the hypothesis that
heterozygous FCHO1 mutants may have a dominant-negative effect on T cell cytokine
production. However, all parents and heterozygous siblings from all six pedigrees were
clinically healthy. We do not have clinical data regarding blood cell counts for parents and/or
heterozygous siblings from this family. Unfortunately, C1 was the only FCHO1-deficient
patient in whom we were able to perform functional experiments in primary T cells. These
cytokine secretion assays are currently the only evidence that heterozygosity might have
some impact on function of peripheral T cells.

Following the reviewer's advice, we discussed this matter in the discussion of revised
manuscript, l.301-305.

*Point 7. Figure S2. The authors state that "point mutation in kindred E affects a splice donor*
*and thereby results in a shortening of the FCHO1 by 51 amino acid encoded by exon 8 (lines*
*143-144). However in panel (e) the differences between healthy donor and patient are not*
*evident. The authors should add a sample from a heterozygous relative to clearly identify the*
*two bands. Also, it is not clear why the E1 sample is run in triplicate. Is it the same sample or*
*different samples from the same patient?*

We are grateful for this comment as it prompted us to look into expression of FCHO1 in
variant E more closely and allowed to critically reassess the protein expression.
WB-detection of the FCHO1 protein is notoriously difficult. The only available (and working)
FCHO1-specific antibodies are polyclonal. In our experience, the specificity varies from lot to
lot. Whereas the assessment of FCHO1 protein expression in immortalized cell lines (using
appropriate positive and negative controls) could be done in a more consistent way, assaying
primary cells has always been challenging and non-specific bands have appeared.
We have done additional experiments and found out that the FCHO1 mutation in kindred E
leads to dual effects: truncation and instability. As pointed out in our original manuscript the
mutation, as documented by Sanger sequencing, results in a deletion of 51 amino acids. The
results of the Sanger sequencing studies are now shown in Sup. Fig. 2c.

Our new data show that the shortened version of the translated FCHO1 protein appears to
be unstable. In our revised paper we show a new Western Blot experiment indicating that the
shortened product cannot be detected by the polyclonal anti-FCHO1 antibodies Sup. Fig. 2d.

Unfortunately, we do not have any fibroblast samples of the heterozygous relative, hence we
could not include it into the analysis.

*Point 8. Discussion. Lines 331-334. The authors list the three possible outcomes of a defect*
*in TCR internalization, including the opposite outcomes of signal amplification or attenuation.*
*They should discuss their results in the context of these possible outcomes.*

The outcome of partially impaired internalisation is essentially unpredictable. We have tried
to address this issue experimentally, using Jurkat cells as a model. Although the data
suggested hypo-phosphorylation of some TCR-associated kinases, the data were not entirely
conclusive. Thus, we decided not to include them into the manuscript. We have added a
short passage into the discussion, please see l. 416-422 of revised manuscript

*Minor points*
*Point 1. Figure 1 is mislabelled. There is no panel (b), at variance with the figure legend. The*
*lettering of the panels should be changed accordingly in the description of the figure in the*
*Results.*

We are not sure what is meant by this comment. Figure 1 contains panel b: Sanger
sequencing chromatograms for families a-d. It is also adequately labelled in the respective
Figure legend.

*Point 2. In all immunoblots please use a line in addition to the molecular mass to precisely*
*indicate the migration of each molecular mass marker*

We appreciate this suggestion and have updated the immunoblot accordingly.

*Point 3. Line 401, legend to figure 2; "In D arrows..." (D) should be in lowercase (d)*

We have corrected this, see line 492.

*Point 4. Figure 4b, line 456. "Unstimulated controls are depicted in black". They are actually*
*depicted in grey (with a black outline)*

Corrected (l. 565).

Point 5. The figures include a figure 6 and a supplemental figure 7 that are not part of the manuscript

We apologize for our carelessness during the final steps of submission. We had deleted Figure 6 prior to submission but an earlier version of the figure had eventually been submitted by mistake – this explains why there is a Figure 6 but neither a legend nor any description in the text.

Point 6. Figure S3. Do fibroblasts lack FCHO1 expression? This should be explicitly stated and the tissue specificity mentioned in the Introduction. Also, as mentioned above, the amplicons should be sequenced to confirm the predictions. In (a) several panels are truncated and they are all very fuzzy. In (e) does HD1/HD2 correspond to ctr1/ctr2 of panel (d)? If so they should have the same label.

All cells, including fibroblasts, express FCHO1 at very low levels. As pointed out above (see response to comment 7), it is possible to detect FCHO1 expression in fibroblasts.

We have sequenced the amplicons, and thus corroborated the notion on splice variant predictions. We apologize for the fuzzy version of the original figure – resulting from multi-author corrections. The new figure (see new Sup. Fig. 3) is in better shape.

Point 7. Figure S6. It is not clear why at longer time points the amount of FCHO1 and EPS1 co-immunoprecipitated with the N-flagged FCHO1 increases while the amount co-immunoprecipitating with the C-flagged FCHO1 decreases. Also for coherence with the other blots all molecular mass markers should be indicated in black.

We have not addressed this issue in great detail. However, our results are consistent with the original observations by Dr. Harvey McMahon, who has characterized the FCHO-proteins (see Henne WM et al., Science 2010, p.1281). These authors have pointed out that N-flagged FCHO1 can be expressed to levels comparable to physiological expression levels, whereas C-flagged FCHO1 appears to be less stable. In our hands, upon stimulation N-flagged FCHO1 co-precipitate more efficiently, possibly due to CD3-induced the clustering. It is well possible, that C-flagged version is functionally impaired or less stable upon stimulation, hence the corresponding signal is fading over time.

Point 9. Line 430: how are "CD3 clusters" defined for the purpose of image analysis? Do they have a min-max size?

"CD3 clusters" have not been defined strictly by size, but rather as structures bigger from the signals observed at time point 0 minutes.

Point 10. Line 183. The authors refer to Sup. Movie 1 (also in the respective legend), but these are actually 3 movies.

The error has been corrected.

**Referee #3: endocytosis**

**Reviewer #3 (Remarks to the Author):**

*The manuscript from Klein and co-workers describes their identification of six (and functional*
*analysis of three) FCHO1 mutations in humans.*

*The authors convincingly show an essential role for FCHO1 in clathrin-mediated endocytosis*
*in T-cell function. This is an important result especially if the improvements mentioned below*
*can be incorporated into the paper.*

*Mutant FCHO1 proteins are affected in the F-BAR or μ HD regions and are thus expected to*
*be compromised in either plasma membrane localisation or in their interaction. with binding*
*partners of FCHO1 μ HD respectively. The μ HD mutants R679P and Stop687 are shown to*
*be severely perturbed in their ability to immuno-precipitate the FCHO1 binding partners*
*EPS15 and EPS15R (Fig 2a). These conclusion are nicely supported.*

We are grateful for the reviewer's general assessment.

**The appearance of GFP-FCHO1-A34P in Fig2d and Sup Fig5 (large aggregates dissociated*
*from the plasma membrane) is not consistent with the appearance of the same protein in Fig*
*2b & Fig 2c (well-defined punctae that appear to co-localise with EPS15 and adaptin,*
*respectively). The reason for this discrepancy is not made clear by the authors.*

The size discrepancy of A34P clusters results from two factors: a) the focus plain chosen
(plasma membrane vs centre of the cell volume) and b) signal intensity as a result of protein
abundance. As the formation of the large clusters (at times enormously large) is a very
unique feature of A34P mutant, we emphasized it in the original version of the manuscript.
However, since it is not the central focus of the paper, and may lead to potential confusion,
we removed the original version of the Sup Fig. 5 and prepared the new Sup. Fig. 5–7. Here,
we address in more fully both distribution and formation of the clusters by all four variants of
FCHO1.

*In agreement with Fig 2a the μ HD mutants R679P and Stop687 do not co-localise with*
*EPS15 and adaptin (Fig 2b, 2c).*

** Worryingly, the images of RFP-clathrin in Fig 2d do not appear to be of acceptable quality*
*to justify interpretations or quantitation (showed in Fig 2e).*

We concur with the reviewer that the strength of the signal coming from endogenously
tagged clathrin light chain A locus in human SK-MEL-2 cells is dim. This cellular system has
been chosen intentionally. One locus of the clathrin A light chain gene has been modified
specifically to yield physiological expression levels of clathrin A. This is critical, since
overexpression of clathrin A would entail unwanted and non-physiological side effects
perturbing the system. We adopted this system from (Doyon, Zeitler et al. 2011), page 333,
left column, second paragraph (reference #15 in our original paper): “Our single-allele
tagged, genome-edited cell lines exhibited considerably dimmer fluorescent structures than
the stable overexpression cell lines. As such, we were concerned that this decreased
fluorescence might affect our ability to faithfully quantify endocytic protein dynamics. To
address this concern, we conducted two tests of the effects of low fluorescence signal on our
lifetime measurements. “

and concludes on the same page:

„These data collectively demonstrate that the observed shorter mean lifetimes of our single-
allele genome-edited lines, compared with all-allele edited and overexpression lines, were
not the result of a decreased ability to detect clathrin for its entire membrane lifetime.”

Page 333, right column, end of paragraph:

„Collectively, these findings demonstrate a more faithful representation of CME than was
achieved using previous methods requiring overexpression and highlight the importance of
studying the dynamics of proteins at physiological expression levels.”

After personal communication with the group of Prof. David Drubin we concluded that we had
to rely on a cellular system with low physiological expression levels of clathrin to study the
effects of FCHO1 on initiation of CME.

To respect the reviewer's concerns, we have performed additional experiments. We stably
overexpressed clathrin light chain A fused to RFP in SK-MEL-2 cells and transiently
transfected them with variants of FCHO1 (Sup. Fig. 7). Next, we quantified the correlation of
RFP to GFP signal. The results obtained from this experiment match the quantification data
in figure 2e. Images are less noisy and red fluorescence is more vivid, however one can also
observe worrying accumulation of RFP signal around the nucleus, which is not the case for
the RFP expressed from single endogenous locus. Furthermore, the cells exhibiting high
expression of tagged-clathrin were notoriously difficult to transfect with FCHO1 constructs.

(Henne, Boucrot et al. 2010) on p.1281 have pointed out that cellular studies on FCHO1
critically depend on their physiological expression levels. In view of these concerns, we
decided to restore clathrin light chain A expression by engineering one allele allowing us to
express a clathrin-RFP fusion protein under the physiological transcription control.

Confronted with the dilemma of either working with low physiological expression levels or
high artificial expression levels, we opted for the scenario that allows us to come up with
meaningful insights based on physiological expression levels.

** The results presented in Fig 3a are also not of a quality to permit interpretation. Neither the*
*movies nor the panels presented as Fig 3a are convincing. The quality of imaging needs to*
*be improved.*

As we have explained above, to maintain physiological levels of clathrin we restored to
single-locus tagged endogenous clathrin light chain. Hence, the overall signal is rather dim.
Please note, however, that despite relatively high noise-to-signal ratio, our conclusion is
based on statistically significant results.

*Thus far, the data show the μ HD mutants R679P and Stop687 are impaired in interaction*
*with binding partners and are thus mis-localised. However, interpretation of the functional*
*outcome of the A34P mutation is not clear.*

We are grateful for this comment. Our data presented in Fig 2 and Fig 3 of the original
manuscript demonstrate that two analysed mutations in the μ HD domain of FCHO1
abolished the interaction of this protein variant with binding partners, EPS15 and EPS15R.
Furthermore, the interaction with adaptin and clathrin is abolished. This indicates that, once

the initial interaction of FCHO1 with EPS15 proteins is perturbed, the entire sequence of
events leading to formation of clathrin coated pit (CCP) is not productive anymore.
Regarding the F-BAR mutant (A34P), our microscopic analysis revealed that this variant form
large protein aggregates (Fig. 2 and Sup. Fig. 5). This is distinct from the small, punctate
structures seen in wt variant. The interaction with the direct binding partners EPS15 and
EPS15R appears to be productive. Nevertheless, the interaction with more distantly
interacting proteins like adaptin or clathrin itself is severely compromised (Fig. 2 and Fig. 3).
Based in this, we conclude that even though aggregates of A34P mutant co-precipitate with
its direct binding partners, EPS15 and EPS15R, massive aggregation led to de-association
of this variant from the plasma membrane preventing productive formation of CCPs.
We have performed additional experiments for the revision of our paper and the new
experimental data support our conclusion: We used fluorescently labelled WGA to visualise
the plasma membrane of SK-MEL-2 cells transiently transfected with FCHO1 expression
constructs (wildtype or mutants). These data demonstrate that: a) overall morphology of cells
expressing mutated versions of FCHO1 is not perturbed; b) F-BAR mutant as well as μ HD
mutants to some extent de-associated from the plasma membrane (Sup. Fig. 5).

*Given that humans harbouring FCHO1 mutations display T cell defects, the authors then*
*examine the possibility that FCHO1 may be involved in internalisation of the CD3:TCR*
*complex. The data presented in Fig 4 are convincing and suggest that FCHO1 indeed plays*
*a role in internalisation of the complex. The authors then go on to examine possible*
*interaction between FCHO1 and CD3 molecules that possess putative μ HD interaction motifs*
*–*
** however, the experiments presented in Sup Fig 6 have not been sufficiently described in*
*the Legend for this reviewer to be able to participate in its interpretation.*

We apologize for any misunderstanding. Figure legend for Sup. Fig. 6 (now Sup. Fig. 9) does
indeed not explain all experimental details. However, all necessary details are provided
elsewhere: the rationale and the experiment has been described in text (lines 238-246 in new
version) and all pertinent information is indicated in MM section (lines 817-844). Should the
reviewer believe that even more information is needed, we are happy to provide the
necessary information.

*Finally, the authors demonstrate that FCHO1 modulates T cell proliferation and cytokine*
*secretion in primary human cells (Fig 5).*
** Note: the submitted manuscript also contains an additional figure (Fig 6) but no mention*
*has been made of this figure in the Legends or in the text.*

We apologize for our carelessness. We had deleted Figure 6 prior to submission but an
earlier version of the paper had eventually been submitted by mistake. This is now corrected.

*In summary, the manuscript should be significantly improved before publication. The*
*improvements suggested above, if incorporated, would make the manuscript suitable for*
*publication because the authors present a very interesting aspect of FCHO1 biology. Also, it*
*would be informative for the authors to comment on why FCHO2 is not able to compensate*
*for the defects reported in this manuscript.*

The authors are grateful for this encouraging statement. We extensively modified the paper,

following the valuable suggestions of all reviewers. We believe that these changes markedly
improved the quality of our manuscript.
We do agree with the reviewer that it would be interesting to know whether FCHO2 could
compensate for the absence of FCHO1. Unfortunately, currently there are no reagents
(antibodies) available which would allow us to stain FCHO2 and see if it is present at all in
the absence of FCHO1. Concerns have already been raised with respect to interpreting the
function of FCHO1/2 whenever the expression levels are not corresponding to physiological
levels (Henne, Boucrot et al. 2010). In view of these concerns, we paid meticulous care to
work with cells in which FCHO1 is expressed in physiological levels.
Studies of FCHO2 would have to follow the same rules. We therefore believe that
overexpression of a tagged-FCHO2 protein may not be the best strategy to come up with
conclusive data. This experiment however does not fit the scope of our revised manuscript.

**Referee #4: rare disease genomics, bioinformatics**

**Reviewer #4 (Remarks to the Author):**

*The manuscript is in general well written. The authors link homozygous deleterious mutations*
*in FCHO1 to T and B cell lymphopenia. In doing so they make a strong case for the*
*involvement of FCHO1 in development and function of T cells. One interesting aspect of the*
*discovery is the diversity of the disruptive mechanism observed across the seven families via*
*missense, nonsense, splice altering and frameshift changes. The authors have functionally*
*validated the impact of half of these mutations. In particular the splice altering mutations are*
*only shown to impact the protein length. Inclusion of the binding properties or sub-cellular*
*localisation assays for these mutants would further improve the study.*

*The inclusion of a negative result in Suppl. Fig. 6 is highly commendable. In general the*
*discussion section provides a fair and balanced overview of the current state of knowledge*
*regarding FCHO1 and the additional gains and future directions defined by the current study.*
*The authors note the severe malformation observed as a result of knockdown experiments in*
*zebrafish and other model organisms in previous studies. They mention potential*
*evolutionary change in functional role of FCHO1 in humans. Although this is plausible, it is*
*not clear if the mutants observed in the current study lead to a complete loss of function.*
*That would provide an alternate hypothesis as well.*

We are grateful for the reviewer's overall positive assessment.

With respect to complete or incomplete loss of function of FCHO1, we would like to point out
that there is a striking and dramatic phenotype of all discovered mutant alleles. We have
found mutations leading to loss of expression (variant E, modified Sup. Fig. 2c and variants
F1 + F2, Sup. Fig. 3f) or massively affecting either interaction with partner proteins or
subcellular localisation.

Regarding the binding properties of the mutants, as documented on Figure 2a, two tested
μ HD domain mutants (R679P (A1) and Stop687 (C1 and D1)) fail to interact with their direct
binding partners Eps15 and Eps15R as compared to WT and F-BAR domain mutant
(immunoblot). Impaired co-localisation with Eps15 as well as alpha-adaptin has been
confirmed by confocal microscopy (see figure 2b, c and e) as compared to WT FCHO1 and
F-BAR domain mutant A34P. Nevertheless, as we have studied interaction of FCHO1
mutants with a limited set of known interacting partners, we cannot rule out that some of the
homozygous mutants display incomplete loss of function.

Regarding subcellular localisation, our new data now included in Sup Fig. 5 show that all
tested mutants (A34P (B1), R679P (A1) and Stop687 (C1 and D1)) de-associate from
plasma membrane (WGA staining) to various extend in contrast to WT FCHO1, which
expression well correlates with plasma membrane marker WGA.

**Minor points**

*Figure 1 panels likely underwent some last minute changes and hence the labels do not align*
*with the references in the manuscript*

We are grateful for this remark and carefully checked the labels of Figure 1 and respective
references in the text. However, in contrast to this comment, we could not detect any
discrepancies in the original version of the manuscript.

Figure 6 legend is missing

We apologize for the mistake. We had deleted Figure 6 prior to submission but an earlier version of the paper had eventually been submitted by mistake – this explains why there is a Figure 6 but neither a legend nor any description in the text.

Whole exome sequencing and variant filtering: details for families E and G seem to be missing. Also it would be helpful to include the polyphen2 prediction for the missense mutations in this section.

We included missing information regarding variant filtering for families E and G in MM section of revised manuscript, l.711-726. We also added results of polyphen2 predictions for the missense mutations in this section, new Table S2.

**References:**

- Alarcon, B., M. Swamy, H. M. van Santen and W. W. Schamel (2006). "T-cell antigen-
receptor stoichiometry: pre-clustering for sensitivity." EMBO Rep **7**(5): 490-495.
- Balagopalan, L., V. A. Barr and L. E. Samelson (2009). "Endocytic events in TCR signaling:
focus on adapters in microclusters." Immunol Rev **232**(1): 84-98.
- Boyer, C., N. Auphan, F. Luton, J. M. Malburet, M. Barad, J. P. Bizozzero, H. Reggio and A.
892 M. Schmitt-Verhulst (1991). "T cell receptor/CD3 complex internalization following activation
of a cytolytic T cell clone: evidence for a protein kinase C-independent staurosporine-
sensitive step." Eur J Immunol **21**(7): 1623-1634.
- Compeer, E. B., F. Kraus, M. Ecker, G. Redpath, M. Amiezer, N. Rother, P. R. Nicovich, N.
Kapoor-Kaushik, Q. Deng, G. P. B. Samson, Z. Yang, J. Lou, M. Carnell, H. Vartoukian, K.
Gaus and J. Rossy (2018). "A mobile endocytic network connects clathrin-independent
receptor endocytosis to recycling and promotes T cell activation." Nat Commun **9**(1): 1597.
- Daecke, J., O. T. Fackler, M. T. Dittmar and H. G. Krausslich (2005). "Involvement of
clathrin-mediated endocytosis in human immunodeficiency virus type 1 entry." J Virol **79**(3):
1581-1594.
- Dergai, M., A. Iershov, O. Novokhatska, S. Pankivskyi and A. Rynditch (2016). "Evolutionary
Changes on the Way to Clathrin-Mediated Endocytosis in Animals." Genome Biol Evol **8**(3):
588-606.
- Dietrich, J., X. Hou, A. M. Wegener and C. Geisler (1994). "CD3 gamma contains a
phosphoserine-dependent di-leucine motif involved in down-regulation of the T cell receptor."
EMBO J **13**(9): 2156-2166.
- Dietrich, J., J. Kastrup, B. L. Nielsen, N. Odum and C. Geisler (1997). "Regulation and
function of the CD3gamma DxxxLL motif: a binding site for adaptor protein-1 and adaptor
protein-2 in vitro." J Cell Biol **138**(2): 271-281.
- Doyon, J. B., B. Zeitler, J. Cheng, A. T. Cheng, J. M. Cherone, Y. Santiago, A. H. Lee, T. D.
Vo, Y. Doyon, J. C. Miller, D. E. Paschon, L. Zhang, E. J. Rebar, P. D. Gregory, F. D. Urnov
and D. G. Drubin (2011). "Rapid and efficient clathrin-mediated endocytosis revealed in
genome-edited mammalian cells." Nat Cell Biol **13**(3): 331-337.
- Henne, W. M., E. Boucrot, M. Meinecke, E. Evergren, Y. Vallis, R. Mittal and H. T. McMahon
(2010). "FCHo proteins are nucleators of clathrin-mediated endocytosis." Science **328**(5983):
1281-1284.
- Hollopeter, G., J. J. Lange, Y. Zhang, T. N. Vu, M. Gu, M. Ailion, E. J. Lambie, B. D.
Slaughter, J. R. Unruh, L. Florens and E. M. Jorgensen (2014). "The membrane-associated
proteins FCHo and SGIP are allosteric activators of the AP2 clathrin adaptor complex." Elife
**3**.
- Liu, A. P., F. Aguet, G. Danuser and S. L. Schmid (2010). "Local clustering of transferrin
receptors promotes clathrin-coated pit initiation." J Cell Biol **191**(7): 1381-1393.
- Mayle, K. M., A. M. Le and D. T. Kamei (2012). "The intracellular trafficking pathway of
transferrin." Biochim Biophys Acta **1820**(3): 264-281.
- McMahon, H. T. and E. Boucrot (2011). "Molecular mechanism and physiological functions of
clathrin-mediated endocytosis." Nat Rev Mol Cell Biol **12**(8): 517-533.
- Miller, K., M. Shipman, I. S. Trowbridge and C. R. Hopkins (1991). "Transferrin receptors
promote the formation of clathrin lattices." Cell **65**(4): 621-632.
- Ohno, H., J. Stewart, M. C. Fournier, H. Bosshart, I. Rhee, S. Miyatake, T. Saito, A.
Gallusser, T. Kirchhausen and J. S. Bonifacino (1995). "Interaction of tyrosine-based sorting
signals with clathrin-associated proteins." Science **269**(5232): 1872-1875.

Pearse, B. M. and M. S. Robinson (1990). "Clathrin, adaptors, and sorting." Annu Rev Cell
Biol **6**: 151-171.

Popov, S., B. Strack, V. Sanchez-Merino, E. Popova, H. Rosin and H. G. Gottlinger (2011).
"Human immunodeficiency virus type 1 and related primate lentiviruses engage clathrin
through Gag-Pol or Gag." J Virol **85**(8): 3792-3801.

Sherman, E., V. Barr and L. E. Samelson (2013). "Super-resolution characterization of TCR-
dependent signaling clusters." Immunol Rev **251**(1): 21-35.

Telerman, A., R. B. Amson, F. Romasco, J. Wybran, P. Galand and R. Mosselmans (1987).
"Internalization of human T lymphocyte receptors." Eur J Immunol **17**(7): 991-997.

Umasankar, P. K., S. Sanker, J. R. Thieman, S. Chakraborty, B. Wendland, M. Tsang and L.
943 M. Traub (2012). "Distinct and separable activities of the endocytic clathrin-coat components
Fcho1/2 and AP-2 in developmental patterning." Nat Cell Biol **14**(5): 488-501.

Wang, L. H., K. G. Rothberg and R. G. Anderson (1993). "Mis-assembly of clathrin lattices on
endosomes reveals a regulatory switch for coated pit formation." J Cell Biol **123**(5): 1107-
1117.

Willox, A. K., Y. M. Sahraoui and S. J. Royle (2014). "Non-specificity of Pitstop 2 in clathrin-
mediated endocytosis." Biol Open **3**(5): 326-331.

Zinchuk, V. and O. Zinchuk (2008). "Quantitative colocalization analysis of confocal
fluorescence microscopy images." Curr Protoc Cell Biol **Chapter 4**: Unit 4 19.

Reviewers' Comments:

Reviewer #1:

Remarks to the Author:

The authors did a fantastic revision work. They have fully answered my comments regarding the fact that clathrin-endocytosis is not impaired in cells KO for FCHO1 or expressing the mutated versions. All other points have been addressed, except regarding Fig. 4a and the cluster/aggregate/intracellular compartments issue.

I have to really disagree here, this is not a semantic issue at all. As rightfully documented by the studies quoted by the authors, clusters are aggregates of proteins within a membrane, which is the plasma membrane most of the time. These clusters, microclusters or even nanoclusters, have been very widely documented in the T cell field. If they do not believe me, the authors can make a quick search for: ("T cell receptor"[title]) and (clustering[title]) on Pubmed and check what these papers talk about.

Typically, Line 204, "intracellular aggregates of CD3 were formed in wt cells" has to be changed (as well as the Y axis label in the bar chart). What the authors see are intracellular compartments that are positive for TCR. This has nothing to do with aggregation. It is the result of an endocytic process that incorporates cell surface TCR into endocytic vesicles, which eventually accumulate in early and recycling endosomes. This is not aggregation.

Besides, regarding Line 225, the Fig. 4 does not allow any visualisation of clusters at the plasma membrane. TCR clusters can be visualised in TIRF or in super-resolution microscopy (see the work of Dustin, Samelson, Gaus or Schütz for instance). I totally understand the authors cannot have access to these approaches, but then they should not discuss the presence or not of TCR clusters at the plasma membrane. This has to be modified.

Finally, there is a more fundamental problem with Fig. 4a. It is that TCR is highly present within intracellular pools, in resting cells and in activated cells (see an excellent recent review, page 2, <https://onlinelibrary.wiley.com/doi/abs/10.1111/imr.12764>). We are not in a situation where most of TCR is at the plasma membrane in resting cells, and moves to intracellular compartments (named aggregate/clusters by the authors) upon T cell activation. Thus, the affirmation as to "all TCR molecules remained at the plasma membrane" (line 225 and 228) has to be tuned down. Moreover, Fig. 4b shows that there is still quite some internalisation of TCR at the same time point, 60 min, which clearly illustrates that not all TCR is stuck at the plasma membrane.

Reviewer #2:

Remarks to the Author:

The authors have addressed satisfactorily all the issues raised in my previous review, with substantial additional experimental evidence, thereby supporting and strengthening the main messages of the manuscript. As commented in my initial review, the data presented in this manuscript are novel and contribute to our understanding of the pathways of vesicular traffic that have emerged as key components of the process of T cell activation. They additionally identify a new disease gene in the as yet large proportion of primary immunodeficiencies of unknown aetiology.

I have however a question regarding the new supplemental figures 7 and 8.

Figure S7. The authors state that only GFP-tagged wild-type FCHO1 co-localizes with clathrin. From the figure shown this is not clear. Maybe for this and the cell expressing GFP-FCHO1 p.A34P, which both display a vesicular GFP pattern, it would be useful to add a blow-up of the region of interest for better visualization.

Figure S8. The pattern of CD3 staining in this figure, which shows Z-stacks for 3D projections to complement the 2D images shown in figure 4a, is different from the one shown in figure 4a. In the latter there is a clear-cut surface pattern for CD3 in wild-type cells at time 0 and an intracellular cluster pattern at 60 min. At variance, very little surface CD3 can be observed for wild-type cells at time 0 in figure S8. In the same figure CD3 appears to form aggregates at time 0 in KO cells, while a surface associated pattern can be clearly observed at the 60 min time point. Can the authors clarify this observation?

Reviewer #4:

Remarks to the Author:

My initial review of the manuscript was quite positive and the impression remains.

I focus my comments as a continuation of my earlier assessment and defer the immunological interpretation to other reviewers with more expertise. Based on the evidence presented in the draft from the seven families as well as public databases (gnomAD, ExAC etc), it is obvious that loss of FCHO1 is highly deleterious. In the current draft the authors make a convincing case for homozygous perturbation in FCHO1's disrupting T cell development via its role in clathrin mediated endocytosis. However the separate role of F-Bar mutations versus uHD mutations is not as convincing. Only one (pA34P) F-Bar mutation is evaluated which is a missense mutation and may not lead to a complete loss of function. This difference in impact may be reflected in the differences observed in Figure 2D and 2E.

Overall the manuscript is presented in a balanced manner with fair representation of both positive and negative results. The authors should either include evaluation of further mutations in F-Bar domain or include this limitation in the draft.

Reviewers' comments:

**Reviewer #1 (Remarks to the Author):**

The authors did a fantastic revision work. They have fully answered my comments regarding
the fact that clathrin-endocytosis is not impaired in cells KO for FCHO1 or expressing the
mutated versions. All other points have been addressed, except regarding Fig. 4a and the
cluster/aggregate/intracellular compartments issue.

We are very glad to hear that Reviewer #1 is satisfied with our work. At this point we would
like to express our gratitude as we think it improved overall quality of the manuscript.

I have to really disagree here, this is not a semantic issue at all. As rightfully documented by
the studies quoted by the authors, clusters are aggregates of proteins within a membrane,
which is the plasma membrane most of the time. These clusters, microclusters or even
nanoclusters, have been very widely documented in the T cell field. If they do not believe me,
the authors can make a quick search for: ("T cell receptor"[title]) and (clustering[title]) on
Pubmed and check what these papers talk about.

Typically, Line 204, "intracellular aggregates of CD3 were formed in wt cells" has to be
changed (as well as the Y axis label in the bar chart). What the authors see are intracellular
compartments that are positive for TCR. This has nothing to do with aggregation. It is the
result of an endocytic process that incorporates cell surface TCR into endocytic vesicles,
which eventually accumulate in early and recycling endosomes. This is not aggregation.

We changed text accordingly to the Reviewer's recommendation. Lines 217-224 reads now
as follow: "After sixty minutes of TCR triggering by an α -CD3 monoclonal Ab, large
intracellular ~~aggregates~~ clusters of CD3 were formed in wt cells, whereas in FCHO1 ko
clones CD3 molecules remained in a non-clustered form ~~on the plasma membrane~~ (Fig. 4a
and Sup. Fig. 8). Knockout clones reconstituted with wt FCHO1 formed large CD3
~~aggregates~~ clusters, essentially indistinguishable from those in wt cells. In contrast, none of
the FCHO1 mutants were able to rescue the phenotype and thus nearly all CD3 molecules
remained non-clustered ~~on the plasma membrane~~ upon TCR-activation (Fig. 4a)."

Besides, regarding **Line 225**, the Fig.4 does not allow any visualisation of clusters at the
plasma membrane. TCR clusters can be visualised in TIRF or in super-resolution
microscopy (see the work of Dustin, Samelson, Gaus or Schütz for instance). I totally
understand the authors cannot have access to these approaches, but then they should not
discuss the presence or not **of TCR clusters** at the plasma membrane. This has to be
modified.

Finally, there is a more fundamental problem with Fig. 4a. It is that TCR is highly present
within intracellular pools, in resting cells and in activated cells (see an excellent recent
review, page 2, <https://onlinelibrary.wiley.com/doi/abs/10.1111/imr.12764>). We are not in a
situation where most of TCR is at the plasma membrane in resting cells, and moves to
intracellular compartments (named aggregate/clusters by the authors) upon T cell activation.

Thus, the affirmation as to "all TCR molecules remained at the plasma membrane" (line 225
and 228) has to be tuned down. Moreover, Fig. 4b shows that there is still quite some
internalisation of TCR at the same time point, 60 min, which clearly illustrates that not all
TCR is stuck at the plasma membrane.

We did not claim (line 223) that: >>"all TCR molecules remained at the plasma membrane<<
but rather: "nearly all CD3 molecules remained non-clustered ~~on the plasma membrane~~
upon TCR-activation". Thus, we deleted the controversial statement: "on the plasma
membrane" and hope that this precision resolves the dispute.

**Reviewer #2 (Remarks to the Author):**

The authors have addressed satisfactorily all the issues raised in my previous review, with
substantial additional experimental evidence, thereby supporting and strengthening the main
messages of the manuscript. As commented in my initial review, the data presented in this
manuscript are novel and contribute to our understanding of the pathways of vesicular traffic
that have emerged as key components of the process of T cell activation. They additionally
identify a new disease gene in the as yet large proportion of primary immunodeficiencies of
unknown aetiology.

We are very glad to hear that Reviewer #2 is satisfied with our responses. We also agree
that new experiments implemented into the manuscript strengthen our claims.

I have however a question regarding the new supplemental figures 7 and 8.

Figure S7. The authors state that only GFP-tagged wild-type FCHO1 co-localizes with
clathrin. From the figure shown this is not clear. Maybe for this and the cell expressing GFP-
FCHO1 p.A34P, which both display a vesicular GFP pattern, it would be useful to add a
blow-up of the region of interest for better visualization.

Thank you for this remark, which gives us an opportunity to comment on an important aspect
of our study. Since the native expression levels of both FCHO1 and clathrin are quite low,
the effects of ectopic expression of tagged-proteins (with unphysiologically high levels of
expression) must be interpreted with caution. An isolated image of a single cell will not
suffice to support or refute our hypothesis. To get more objective and interpretable data, we
performed statistical analysis of many images coming from different experiments. In the
main figure, we enlarged the region of interest. To provide additional data, we added a figure
supplementary figure showing a complementary visualisation approach.

Figure S8. The pattern of CD3 staining in this figure, which shows Z-stacks for 3D
projections to complement the 2D images shown in figure 4a, is different from the one shown
in figure 4a. In the latter there is a clear-cut surface pattern for CD3 in wild-type cells at time
0 and an intracellular cluster pattern at 60 min. At variance, very little surface CD3 can be
observed for wild-type cells at time 0 in figure S8. In the same figure CD3 appears to form
aggregates at time 0 in KO cells, while a surface associated pattern can be clearly observed
at the 60 min time point. Can the authors clarify this observation?

We concur with the Reviewer that staining pattern on Fig. 4 and Sup Fig. 8 differs to some
extent. It can be in part attributed to the fact that both experiments were analysed on two
different microscopes, which differs in terms of sensitivity. In the main figure 4, superior
resolution prevents us from detection of small CD3 clusters which are visible both in WT and
KO cells in Sup Fig 8. The main message, however, remains unchanged.

In WT scenario at time point 0, surface CD3 molecules (top panels) are visible, but rather
dim. Here, some intracellular CD3 clusters can be observed, although not very big and
bright. At 60 min after stimulation the situation changes, as almost all CD3 molecules are
concentrated intracellularly forming large, bright aggregates. KO cells, in contrast to WT cells
do not change their status of CD3 expression. At both time points (0 min and 60 min after
stimulation) CD3 molecules are located partially on the cell surface and form rather small
clusters (at least when compared to that of WT 60 min.).

In experiment shown in Fig 4 we could comment only on large, vivid clusters formed 60 min
upon stimulation, where the status of WT and KO cells is clearly different.

Reviewer #4 (Remarks to the Author):

My initial review of the manuscript was quite positive and the impression remains.

We are very pleased to read this comment.

I focus my comments as a continuation of my earlier assessment and defer the
immunological interpretation to other reviewers with more expertise. Based on the evidence
presented in the draft from the seven families as well as public databases (gnomAD, ExAC
etc), it is obvious that loss of FCHO1 is highly deleterious. In the current draft the authors
make a convincing case for homozygous perturbation in FCHO1's disrupting T cell
development via its role in clathrin mediated endocytosis. However, the separate role of F-
Bar mutations versus uHD mutations is not as convincing. Only one (pA34P) F-Bar mutation
is evaluated which is a missense mutation and may not lead to a complete loss of function.
This difference in impact may be reflected in the differences observed in Figure 2D and 2E.

Overall the manuscript is presented in a balanced manner with fair representation of both
positive and negative results. The authors should either include evaluation of further
mutations in F-Bar domain or include this limitation in the draft.

Please note, that we have identified three distinct mutations in the F-BAR domain, all of
which have been characterised and discussed in the manuscript. Two of those (kindred E
and F) are resulting in very short and unstable protein, and *de facto* could be considered as
null allele. As the results of these mutations are rather predictable, results are shown in Sup.
Fig. 2 and Sup. Fig. 3.

In contrast to the truncating mutations, the consequences of the point mutations (both in F-
BAR and uHD) are far less obvious (and probably far more interesting for the readers),
hence they were characterised in much greater detail. The results were put into main body of
the manuscript. Here, we show mechanistic impact of both point mutations and explain why
both are deleterious for FCHO1 function. As shown in Fig. 2 and Sup. Fig 5, mutation in F-
BAR domain does not affect interaction with binding partners but alters cellular localisation of
FCHO1 (particularly well illustrated in Sup. Fig. 5). In contrary, mutations in uHD domain

prevent FCHO1 from interaction with key binding partners (Fig. 2a and Fig. 2e). Although the
molecular basis differs, the functional consequences are the same – neither of the FCHO1
variants is able to promote nucleation of clathrin-coated pits.

**Reviewer #3 failed to deliver a report and we asked Reviewer #2 to comment on the**
**points raised. Please find below their feedback to us:**

"Overall, the authors have addressed satisfactorily the issues raised by reviewer 3. Two
minor points, highlighted in the attached file, should be addressed in the Discussion:

Point 1. While the authors argue in a convincing fashion the functional outcome of the A34P
mutation in their response to the reviewer, this is not commented upon in the text. A short
sentence to comment this point should be added in the Discussion

*We actually did comment on this in our text. Lines 360 – 363 read: “Alterations in the F-BAR*
*domain also prove to be deleterious, yet in a different mode. Substitution of Ala at position*
*34 with Pro, a known alpha-helix breaker, leads to dissociation of the FCHO1 homodimers*
*from the plasma membrane and hence abolishes the function.” Here we do not mention*
*about the size of the aggregates of A34P mutant as it is not a main factor determining its*
*deleteriousness.*

Point 2. The authors should comment on why FCHO2 is not able to compensate for the
defects reported in the manuscript. The authors state that it is not known whether FCHO2 is
expressed in the absence of FCHO1 but that they cannot check this due to the lack of a
good antibody. A qRT-PCR-based measurement of the transcript would provide some
preliminary information. However, even in the absence of additional experiments which are
somewhat out of the scope of this manuscript, they could add a comment, underscoring the
caveat about the possible effect of FCHO1 deficiency on FCHO2 expression."

*We concur with the Reviewer that compensation of the FCHO1 defect by its paralogue*
*FCHO2 is a very interesting subject which deserve in-depth investigation. However, we also*
*believe that this, in the absence of additional data, would rather distract from the main*
*subject of understanding the mechanism underpinning FCHO1-mediated immunodeficiency.*
*There are at least two plausible reasons, why FCHO2 cannot replace FCHO1:*
*1/ although both proteins share the same F-BAR and μ HD domains, there is only limited*
*conservancy between them, suggesting that precise function of these proteins may differ.*
*2/ The functional discrepancies between the two paralogues has been documented for the*
*less organised linker between the two domains (a.a. 267-467). The central segment of*
*FCHO1 is responsible for direct interaction with the AP-2 complex, while FCHO2 does not*
*associate with AP-2 (P.K. Umasankar et al, Nat Cell Bio, 2012)*

*We appreciate that Reviewer #3 invites as to add a comment (“they could add a comment”),*
*but we would like to ask the editor for permission to refrain from this in the absence of*
*additional data. We feel more comfortable not to speculate too much.*

Reviewer #2 also provided a document to highlight which points they are referring to and I
have attached this document here. Please include these revisions in your revised
manuscript.

*All those highlighted points are commented above.*